# Task Planning for Visual Room Rearrangement under Partial Observability

**Karan Mirakhor\*, Sourav Ghosh\*, Dipanjan Das & Brojeshwar Bhowmick**
Visual Computing and Embodied Intelligence Lab
TCS Research, Kolkata, India
`{karan.mirakhor, g.sourav10, dipanjan.da, b.bhowmick}@tcs.com`

## Abstract

This paper presents a novel modular task planner under partial observability that empowers an embodied agent to use visual input to efficiently plan a sequence of actions for simultaneous object search and rearrangement in an untidy room, to achieve a desired tidy state. The paper introduces **(i)** a novel Search Network that utilizes commonsense knowledge from large language models to find unseen objects, **(ii)** a Deep RL network trained with proxy reward, along with **(iii)** a novel graph-based state representation to produce a scalable and effective planner that interleaves object search and rearrangement to minimize the number of steps taken and overall traversal of the agent, as well as to resolve blocked goal and swap cases, and **(iv)** a sample-efficient cluster-biased sampling for simultaneous training of the proxy reward network along with the Deep RL network. Furthermore, the paper presents new metrics and a benchmark dataset - RoPOR, to measure the effectiveness of rearrangement planning. Experimental results show that our method significantly outperforms the state-of-the-art rearrangement methodsWeihs et al. (2021); Gadre et al. (2022); Sarch et al. (2022); Ghosh et al. (2022).

## 1 Introduction

Tidying a disordered room based on user specifications is a challenging task as it involves addressing issues related to perception, planning, navigation, and manipulation Batra et al. (2020). An agent performing an embodied room rearrangement must use the sensor observations and a prior knowledge to produce a long horizon plan for generating a sequence of object movements to achieve the tidy goal state. This goal state is specified through geometry, images, language, etc. Batra et al. (2020).

Majority of the existing research on room rearrangement emphasizes on perception and commonsense reasoning while assuming navigation and manipulation abilities, without incorporating efficient planning. Based on the goal state definition, they broadly fall into two categories; (i) *Commonsense based reasoning without a predefined goal state*: The methods Kant et al. (2022); Sarch et al. (2022) in this category utilize image or language-based commonsense reasoning to identify if an object is misplaced from the correct receptacles in their egoview followed by rearranging them using a sub-optimal heuristic planner. Moreover, utilizing text or semantic relation-based anomaly detectors to identify misplaced objects does not resolve blocked goal or swap cases, where an object's goal position is obstructed by another misplaced object or vice versa. (ii) *User-specific room rearrangement with a pre-defined tidy goal state*: In this setting, the rearrangement is done based on explicit user specification. Methods like Weihs et al. (2021); Gadre et al. (2022) focus on egocentric perception and use image or image feature-based scene representation to identify misplaced objects and a greedy planner to sequence actions for rearrangement. Sarch et al. (2022) also performs a user-specific room rearrangement by using semantic relations to identify misplaced objects in agent's egoview, and then rearrange them as they appear without planning. Methods such as Kant et al. (2022); Sarch et al. (2022); Gadre et al. (2022) explicitly explore the room to find objects that are initially outside the agent's egoview, since it only provides a partial information about the room. However, these approaches incur a significant traversal cost due to exploration. Additionally, these methods employ non-optimal planning that does not optimize the number of steps or overall traversal.

In contrast, efficient planning makes rearrangement more effective by optimizing the sequence of actions and minimizing the time and effort required to achieve the goal state. Ghosh et al. (2022),

---

\*These authors contributed equally.

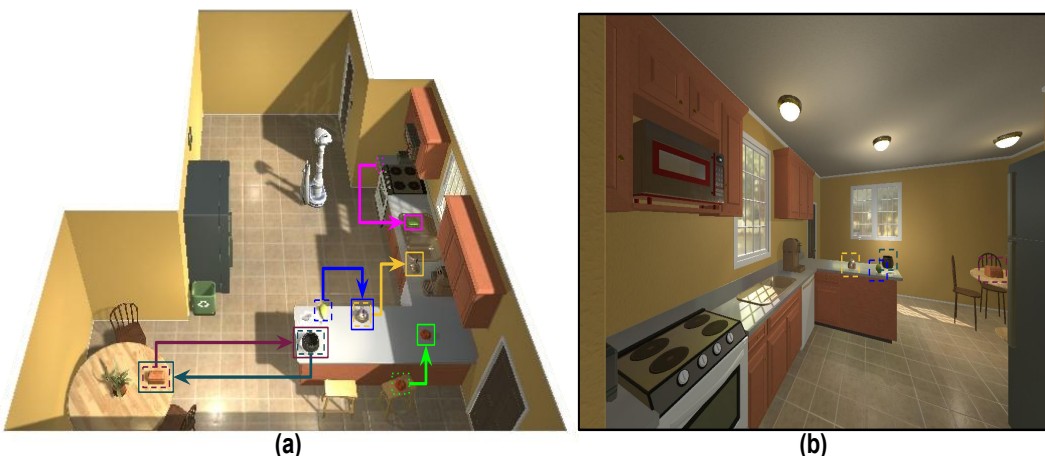

Figure 1: **(a)** shows the top down view of our Rearrangement task and **(b)** is the agent's initial egocentric view in the untidy current state for the same setup. The solid 2D bounding boxes indicate the desired goal state for all objects, while the dashed ones show the initial positions of visible objects in the untidy current state. The dotted 2D bounding boxes represent initial positions of unseen objects in the untidy current state. The sponge (magenta), an unseen object, is in a drawer near the stove, while the tomato (green), another unseen object, is on a stool behind the countertop. There are two scenarios: a blocked goal case with the lettuce (blue) and kettle (yellow) and a swap case between the bread (dark magenta) and pot (dark cyan).

addresses the rearrangement task planning problem by assuming the complete visibility of the room, through the bird's eye view. Their method addresses some planning problems, such as the combinatorial expansion of rearrangement sequencing, and blocked goal and swap cases without explicit buffer. However, the approach does not minimize overall agent traversal during the planning, and its state representation is not scalable to large numbers of objects. Moreover, their reliance on the ground truth object positions in both the current and goal states is impractical in real-life. Our aim is directed towards a novel and more practical aspect of the room rearrangement problem through efficient task planning under partial observability of a room using agent's egocentric camera view.

The major challenges associated with efficient task planning for room rearrangement under partial observability, as shown in Fig. 1, are **(i)** uncertainty over the location of unseen objects due to partial observability (objects outside the agent's field of view presently which are visible from a different perspective, or objects placed within a closed receptacle e.g. spoon in drawer), **(ii)** scalability to a large number of objects, **(iii)** combinatorial expansion of sequencing due to simultaneous object search (for unseen objects) and rearrangement, **(iv)** minimizing the overall traversal during simultaneous object search and rearrangement, **(v)** blocked goal and swap cases without explicit buffer.

In this paper, we propose a *novel modular method* for a task planner to address the aforementioned challenges. At the beginning, our agent captures the goal state by exploring the room to record the semantic and the geometric configuration Batra et al. (2020) of objects and receptacles through egocentric perception. Once the goal state is captured, the objects in the room are shuffled. In the untidy current state, our method partitions the task planning problem into two parts; object search and planning, with the aim of minimizing the overall agent traversal during simultaneous object search and rearrangement. **First**, we propose a novel commonsense knowledge based Search Network using large language models (LLMs) Liu et al. (2019); Kant et al. (2022) that leverages the object-receptacle semantics to predict the most probable receptacle for an unseen object in the egoview. **Second**, we use a Deep RL network with hybrid action space Ghosh et al. (2022) to plan our action sequence for simultaneous object search and rearrangement by resolving blocked goal and the swap cases. To this extent, we define the Deep RL state space with a novel graph-based state representation for the current and the goal state that incorporates geometric information about objects. This representation compactly encodes the scene geometry that aids in rearrangement planning and makes the Deep RL state space scalable to a large number of objects and scene invariant. In addition, we present a novel, sample-efficient cluster-biased sampling for simultaneous training of the proxy reward network Ren et al. (2022) and Deep RL to get a better estimate of the problem's true objective from the episodic reward than the dense reward in Ghosh et al. (2022). The judicious combination of all the

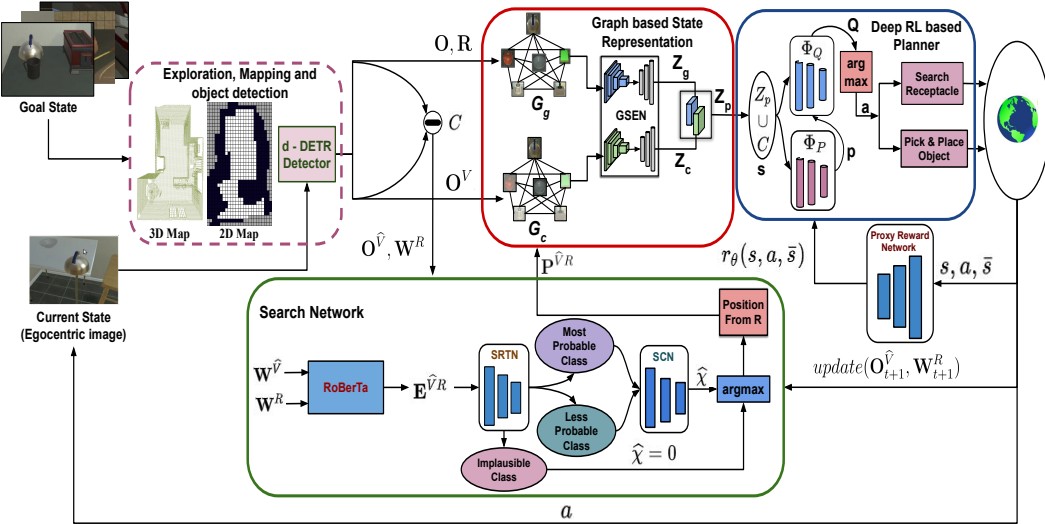

Figure 2: Overall pipeline of our proposed method.

aforementioned components effectively tackle the challenging combinatorial optimization problem in rearrangement as detailed in Sec. 3.6.

The major contributions of this paper are :

1. To the best of our knowledge, this is the **first end-to-end method** to address the task planning problem for room-rearrangement from an egocentric view under partial observability, using a user-defined goal state.
2. A novel **Search Network** that leverages object-receptacle semantics using the commonsense knowledge from LLMs to predict the most probable receptacle for an unseen object.
3. Use of **Deep RL based planner trained with proxy reward** to overcome combinatorial expansion in rearrangement sequencing and, to optimize the overall traversal and the number of steps taken.
4. A new **Graph-based state representation** for the current and goal state to include geometric information about objects, making the Deep RL state space scalable to large numbers of objects and scene-invariant.
5. Introduction of a novel, sample-efficient **cluster-biased sampling** for simultaneous training of the proxy reward network and the Deep RL network.
6. We introduce a **new set of metrics** in Sec. 3.4 to obtain a thorough assessment of the rearrangement planner's effectiveness by not only evaluating the success of the rearrangement, but also taking into account the number of steps taken and the overall agent traversal.
7. To address the inadequacies in existing benchmarks Weihs et al. (2021) for evaluating task planning under partial observability, we introduce the **RoPOR - Benchmark Dataset**. We plan to openly release the dataset to enable further research in this domain.

## 2 METHODOLOGY

In our room-rearrangement setup, the agent explores the room to capture the tidy user-specified goal state. During this exploration, the agent creates a 2D occupancy map $\mathbf{M}^{2D}$ for the agent's navigation while, 3D map $\mathbf{M}^{3D}$ is utilized to augment the detection of 3D object and receptacle centroids to a fixed global reference frame ($\mathbb{R}^3$). Additionally, we generate an object list $\mathbf{O} = \{[\mathbf{W}_i, \mathbf{P}_i], i = 1, 2, .., N\}$ and a receptacle list $\mathbf{R} = \{[\mathbf{W}_i^R, \mathbf{P}_i^R], i = 1, 2, .., N_R\}$. Here, $N$, $\mathbf{W}$ and $\mathbf{P} \in \mathbb{R}^3$ are the total numbers of objects, their semantic labels, and 3D object centroids, respectively. While $N_R$, $\mathbf{W}^R$ and $\mathbf{P}^R \in \mathbb{R}^3$ are the total numbers of receptacles, their semantic labels including the room name from Ai2Thor Kolve et al. (2017), and the 3D receptacle centroids respectively. Then, we randomly shuffle a few objects from the goal state to make the room untidy and fork the agent at a random location in the room. In this untidy current state, the agent's knowledge is limited to the visible part of the room in its egocentric view. In the agent's egocentric perception, only a set of objects $\mathbf{O}^V = \{[\mathbf{W}_i^V, \mathbf{P}_i^V], i = 1, 2, .., N_V\}$ are visible. $N_V$, $\mathbf{W}^V$ and $\mathbf{P}^V \in \mathbb{R}^3$ are the number of visible objects, their semantic labels, and their 3D object centroids respectively in the current state. Comparing $\mathbf{O}$ in the goal state with $\mathbf{O}^V$ in the current state allows for determining only the semantics

of unseen objects $\mathbf{O}^{\widehat{V}} = \{\mathbf{W}_i^{\widehat{V}}, i = 1, 2, .., N_{\widehat{V}}\}$, where $N_{\widehat{V}}$ is the number of unseen objects and $\mathbf{W}^{\widehat{V}}$ their semantic labels. To plan efficiently and achieve the goal state, the agent must know the positions of all objects in the current state. This involves optimizing the search for unseen objects based on the object-receptacle semantics and simultaneously rearranging visible objects based on their positions in the current and goal state. To this end, we present a modular approach for task planner, as shown in Fig. 2, with : **(i)** *Search network*, **(ii)** *Graph-based state representation*, **(iii)** *Deep RL network trained with proxy reward*. The objective of our task planner is to minimize the number of steps and the agent's overall traversal by simultaneously sequencing high-level actions to either *pick-place* misplaced objects or *search* for unseen objects at predicted receptacles.

## 2.1 BACKGROUND

The agent maps the room in the goal state using an exploration strategy Sarch et al. (2022) and receives RGB-D images and egomotion information at each step from Ai2Thor Kolve et al. (2017). The agent constructs $\mathbf{M}^{2D}$ and $\mathbf{M}^{3D}$ of the environment using the RGB-D input and egomotion. A d-DETR Zhu et al. (2021) detector is used on the RGB images to obtain 2D bounding boxes and semantic labels for objects and receptacles, and the corresponding 3D centroids are obtained using depth input, camera intrinsic and extrinsic. Finally, the agent has $\mathbf{O}$, $\mathbf{R}$, $\mathbf{M}^{2D}$, and $\mathbf{M}^{3D}$ from the goal state. In the current state, the agent uses d-DETR detector Zhu et al. (2021) along with $\mathbf{M}^{3D}$ to obtain $\mathbf{O}^V$. The agent uses the *Djikstra* path planner on $\mathbf{M}^{2D}$ to navigate and execute high-level actions by assuming perfect motion and manipulation capabilities.

## 2.2 SEARCH NETWORK

We present a novel LLM-based *Search Network* to reliably predict the receptacles for $\mathbf{O}^{\widehat{V}}$. In case the predicted receptacle is articulated, the agent opens it and looks for the object. The agent uses the predicted receptacle's position from the goal state to be the probable location for $\mathbf{O}^{\widehat{V}}$ in the current state, since receptacles are static in the room. To this end, we finetune the RoBERTa embeddings to exploit the commonsense knowledge in LLM and learn the semantic relationship between $\mathbf{O}^{\widehat{V}}$ and $\mathbf{R}$. Fine-tuning LLM embeddings is essential because LLMs, being trained on large data corpus, may not necessarily produce human-commonsense compliant predictions for untidy scenes (see the Appendix for more details). Our Search Network (SN) consists of two parts: the Sorting Network (SRTN) and the Scoring Network (SCN). We use RoBERTa-Large model Liu et al. (2019) to generate pairwise embeddings ($\mathbf{E}^{\widehat{V}R}$) for $\{\mathbf{W}_i^{\widehat{V}}\}_{i=1,2,..,N_{\widehat{V}}}$ and $\{\mathbf{W}_i^R\}_{i=1,2,..,N_R}$ in the current state. Therefore, there are $N_E = N_{\widehat{V}} \times N_R$ number of embeddings for all the object-room-receptacle (ORR) pairs. Each ORR embedding is classified into one of the 3 classes, based on the probability $\{p_i\}_{i=1,2,3}$ from the Sorting Network. The ground truth class labels $\{Y_i\}_{i=1,2,3}$ for each ORR in the dataset (Sec. 3.1) is based on the probability to find an object at that room-receptacle, where $\{i = 1 : $ *Most Probable Class*, $2 : $ *Less Probable Class*, $3 : $ *Implausible Class*$\}$. SRTN filters out the room-receptacles, where there is a negligible chance of finding the misplaced object. For instance, even in an untidy room, it is nearly impossible to find a cup in the bathtub of a bathroom. This sorting step reduces the scoring network's computation and minimizes the chances of erroneous scoring of an implausible ORR. We train a fully connected MLP in SRTN using the Cross-Entropy Loss ($L_{CE}$) as shown in Eq. (1). The Scoring Network estimates probability scores $\{\widehat{\chi}_i\}_{i=1,2,...N_{SR}}$ for embeddings of higher probability classes, with $N_{SR}$ representing the total number of such embeddings. SCN provides a probability score metric, to choose the most probable receptacle for $O^{\widehat{V}}$. For training the fully connected MLP in SCN, we calculate the MSE Loss ($L_{MSE}$) of probability scores, as in Eq. (2), with respect to the ground truth probability scores $\{\chi_i\}_{i=1,..N_{SR}}$. Finally, we get the position ($\{\mathbf{P}_i^{\widehat{V}R}\}_{i=1,..N_{\widehat{V}}}$) of the unseen objects as the position of their most probable receptacle.

$$L_{CE} = -\frac{1}{N_E} \sum_{i=1}^{N_E} \sum_{i=1}^{3} Y_i \log p_i \tag{1}$$

$$L_{MSE} = \frac{1}{N_{SR}} \sum_{i=1}^{N_{SR}} (\widehat{\chi}_i - \chi_i)^2 \tag{2}$$

To prevent fruitless searches, we implement simple strategies. If the agent cannot find the unseen object at the predicted receptacle, the Search Network identifies the next most probable room-receptacle, and the prior prediction is discarded before re-planning a new sequence. Additionally, if

the agent encounters a receptacle on its path that does not contain any unseen objects, it is removed from future searches. The agent updates $\mathbf{O}^{\widehat{V}}$ whenever it detects an unseen object in its egoview. If the agent locates the unseen object it is searching for before arriving at the predicted receptacle, it updates $\mathbf{O}^{\widehat{V}}$ and re-plans a new sequence. Refer appendix for more details on the re-planning strategy.

## 2.3 GRAPH-BASED STATE REPRESENTATION

For our task planning algorithm, we create a spatial graph ($G = \{V, E\}$) representation of the current and the goal state namely $G_c = \{V_c, E_c\}$ and $G_g = \{V_g, E_g\}$ respectively. The nodes $V_c = \{O^V\}$ and $V_g = \{O\}$. The fully connected edges of the graph contain the path length as edge features, where $E_c = \{\mathcal{D}(P_i^V, P_j^V)_{i \neq j}\}$ and $E_g = \{\mathcal{D}(P_i, P_j)_{i \neq j}\}$. The path length $\mathcal{D}(A_i, A_j)_{i \neq j}$ is the length of the shortest collision free path, computed using $Djikstra$, between the 2D projections of $A_i, A_j \in \mathbb{R}^3$ on $\mathbf{M}^{2D}$. For unseen objects in the current state, the object nodes and edges in $G_c$ are augmented with $P^{\widehat{V}R}$ from the search network as $V_c = V_c \cup \{O^{\widehat{V}}, P^{\widehat{V}R}\}$ and $E_c = \{\mathcal{D}(\overline{P}_i, \overline{P}_j)_{i \neq j}\}$, where $\overline{P} = P^V \cup P^{\widehat{V}R}$. This graph representation helps the Deep RL state space to understand the semantic and geometric information of the current and the goal state. We use a novel *Graph Representation Network (GRN)* with an encoder-decoder to generate meaningful embeddings from $G_c$ and $G_g$ for Deep RL state space to incorporate the residual relative path length notion between every pair of current and goal state nodes. GRN consists of two major blocks, the Graph Siamese Encoder Network (GSEN) and the Residual Geodesic Distance Network (RGDN) . GSEN uses a Graph Convolution Network (Gao et al., 2020) to encode the graphs $G_c$ and $G_g$ and produce the graph embeddings $Z_c$ and $Z_g$ respectively. These graph embeddings are concatenated to get the final embeddings $Z_p = Z_c \cup Z_g$. RGDN acts as a decoder and predicts the residual relative path length $\tau_p$ between the two graphs. This network is trained in a supervised way as in Eq. (3), using the *Graph Dataset* (Sec. 3.1), which contains the ground truth relative path length ($\tau$) between the two graphs. This graph embedding makes the Deep RL state space invariant to a large number of objects and the scene. This compact representation concisely encodes the pairwise distance between the source and target nodes which aids in the reduction of the combinatorial expansion of rearrangement sequencing.

$$\tau_p = GRN(G_c, G_g)$$
$$L_{GRN} = ||\tau - \tau_p||^2 \tag{3}$$

## 2.4 DEEP RL BASED PLANNER

Our task planner needs to select the objects or the probable receptacles for the unseen objects in an efficient manner, to minimize the overall traversal of the agent to simultaneously search the unseen objects and rearrange the visible ones. Moreover, the planner needs to identify free locations, when selecting objects with swap cases.

### 2.4.1 PARAMETERIZED DEEP-Q NETWORK

In order to achieve the aforementioned goals, we implement a Parameterized Deep-Q Network with hybrid action space, similar to Ghosh et al. (2022). We define a binary Collision vector ($C_{N \times 1}$), that signifies the objects with a blocked goal or swap case. The Deep RL state space defined as $s = Z_p \cup C$. Each action $\{a_i = (k, p_k)\}$ in our sequence of actions $\{\mathbf{a}_i\}_{i=1,2,..,K}$ of length $K$ is made up of a discrete action $k$, denoting the index of the selected object or the probable receptacle, followed by a continuous parameter $p_k$ which signifies the location for object placement or receptacle search. We use a Parameter network ($\Phi_P$) and the Q-network ($\Phi_Q$) to generate a continuous parameter $p_k$ and a discrete action $k$ respectively, similar to Ghosh *et al.*. According to a *Markov Decision Process (MDP)*, our method receives a reward $r(s, a)$ at each time step $t$, for choosing an action $a$, that advances the agent from the current state $s$ to the next state $\bar{s}$. Inspired by the work in Ghosh et al. (2022); Bester et al. (2019), we define the Q-values as a function of the joint continuous action parameter $p = [p_k]_{k=1,2,..,K}$ instead of updating the Q-values with its corresponding continuous parameter sample $p_k$. The modified Bellman equation is shown in Eq. (4). This prevents our method from producing degenerate solutions by incorporating the effect of other parameters for updating the Q-values.

$$Q(s, k, p) = \underset{r, \bar{s}}{E}[r + \gamma \underset{\bar{k} \in K}{max} Q(\bar{s}, \bar{k}, \Phi_P(\bar{s}))|s, k, p] \tag{4}$$

The loss function $L_p(\Phi_P)$ and $L_Q(\Phi_Q)$ for the parameter network($\Phi_P$) and the Q network($\Phi_Q$), is given by Eq. (5)

$$L_P(\Phi_P) = -\sum^{R_B}\sum^{K}_{k=1} Q(s, k, \Phi_P(s); \Phi_Q)$$

$$L_Q(\Phi_Q) = \underset{(s,k,p,r,\bar{s}) \leftarrow R_B}{E} [\frac{1}{2}(y - Q(s, k, p; \Phi_Q))^2]$$

(5)

Here, $y = r + \gamma \underset{k \in K}{max} Q(\bar{s}, \bar{k}, p(\bar{s}; \Phi_P); \Phi_Q)$ is the updated target from Eq. (4) and $R_B$ is the replay buffer. $L_P(\Phi_P)$ indicates how the $p$ must be updated to increase the Q-values. Here $\Phi_Q$ works as critic to $\Phi_P$.

For Long Horizon planning, the sparse reward is not sampling efficient for training the Deep RL Gehring et al. (2021). Hence, we use step-wise environmental feedback based on the hierarchical dense reward similar to Ghosh *et al.*. The detailed reward structure is explained in the Appendix. This reward structure provides per-step feedback, but we need episodic reward-based feedback to improve RL policy generalization Amodei et al. (2016); Dewey (2014). Thus, for every episode ($\Lambda$), we calculate the episodic reward ($R_{ep}$) using the step-wise hierarchical dense reward ($r$) and overall episodic path length ($L$) as in Eq. (6), and save the reward and each step ($s, a, \bar{s}$) of the episode into the replay buffer ($R_B$). As this episodic reward is sparse, we use a proxy reward network to generate per-step dense Markovian reward with an episodic notion.

### 2.4.2 PROXY REWARD NETWORK

Our proxy reward network is trained on the sampled experience data from the replay buffer, to give our agent a notion of the overall objective of the episode. The random return decomposition (RRD) method used in Ren et al. (2022), trains a proxy reward network by randomly sampling steps from an episode. This training method is not sample efficient because it uniformly samples the steps without considering the reward distribution in the episode.

To this end, we propose a novel cluster-biased return reward decomposition (CB-RD) to train our proxy reward network. We cluster the per-step reward for the episode into 3 clusters each of size $T_j$, where $j \in \{1, 2, 3\}$, using the *c-means clustering*. These clusters represent the reward distribution in an episode. This information helps us to efficiently sample $N_s$ number of steps from the episode. We randomly sample $U_j = \{(s_{ij}, a_{ij}, \overline{s_{ij}})\}^{N_j}_{i=1}$ from each cluster $j$, such that $N_j = N_s \times T_j/N_{ep}$. Using $\{U_j\}_{j=1,2,3}$, we estimate the learned episodic reward ($R_{ep,\theta}$) from the proxy reward network ($r_\theta(s, a, \bar{s})$), where $\theta$ is the learned weight.

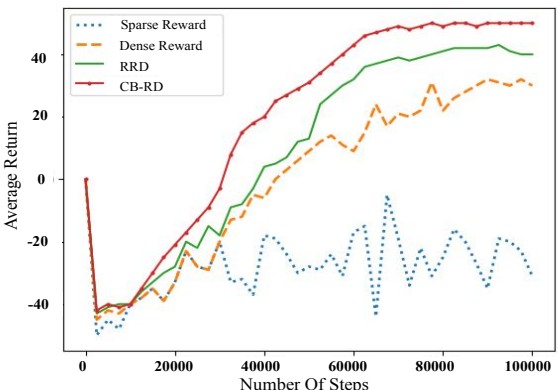

Figure 3: The study compares various reward methods used to train our Deep RL network and finds that our proposed method, CB-RD, achieves the highest average return in a shorter training time, hence improving sampling efficiency.

$$R_{ep} = \frac{N_{ep}}{L}\sum^{N_{ep}}_{i=1} r_i \quad (6)$$

$$R_{ep,\theta} = \sum^{3}_{j=1} p_j \frac{T_j}{N_j}\sum^{N_j}_{i=1} r_\theta(s_{i,j}, a_{i,j}, \overline{s_{i,j}})$$

(7)

$$L_{CBRD} = \frac{1}{M}\sum^{M}_{i=1}\left[(R_{ep_i} - R_{ep,\theta_i})^2\right] \quad (8)$$

Here, $M$ is the number of episodes sampled, $N_{ep}$ is the number of steps in an episode and $p_j = T_j/N_{ep}$ is the uniform probability of choosing a sample from the episode that belongs to cluster $j$. We simultaneously train our Deep RL using Eq. (5) and proxy reward network using Eq. (8) as shown in Algorithm 1. Fig. 3 shows that CB-RD provides effective feedback to our Deep RL method to achieve a higher average return in a lesser number of steps during training. Hence, CB-RD makes

our Deep RL method more sample efficient compared to RRD, hierarchical dense reward and sparse reward.

We use an off-policy method with a replay buffer to train our Deep RL method with a diverse set of rearrangement configurations, similar to the work proposed by Kalashnikov et al. (2018). We use the $\epsilon$ greedy method Kalashnikov et al. (2018) to strike a balance between exploration and exploitation. We stabilize our Deep RL training using target networks for $\Phi_Q$ and $\Phi_p$, and update the weights of target networks using polyak Lillicrap et al. (2015) averaging similar to Bester et al. (2019); Ghosh et al. (2022). Our ablation study in Appendix, shows that the selection of $\epsilon$ has a significant impact on the solution.

## 3 EXPERIMENTS

In this section, we describe the datasets, metrics, and detailed results of our proposed method and its modules, in addressing the room-rearrangement problem.

### 3.1 DATASET

**Graph Dataset :** We generate this dataset to train GRN using Ai2Thor Kolve et al. (2017), by randomly placing objects for two types of rearrangement scenarios: (i) $40\%$ without goal occupied rearrangement: by placing the objects in free spaces and (ii) goal occupied rearrangement: by placing the object in another object's target.

---

**Algorithm 1:** Training Proxy Reward Network

**1** **Initialize** $R_B \leftarrow \{\ \}$;
**2** **for** $i \leftarrow 0, 1, 2, ...$ **do**
**3** $\quad$ Using $\epsilon$-greedy , we rollout episode $\Lambda_i$ ;
**4** $\quad$ Calculate episodic reward $R_{ep_i}$ using $Eq.$ (6);
$\quad\quad R_B \leftarrow R_B \cup \{\Lambda_i, R_{ep_i}\}$;
**5** $\quad$ **for** $l \leftarrow 0, 1, 2, ...$ **do**
**6** $\quad\quad$ Sample M episodes $\{\Lambda_l \in R_B\}_{l=1}^M$;
**7** $\quad\quad$ Group steps from each episode into $\quad\quad j \in \{1, 2, 3\}$ clusters based on r ;
**8** $\quad\quad$ Cluster Biased Sampling to get $U_{lj}$ from $\Lambda_l$;
**9** $\quad\quad$ Compute $R_{ep,\theta}$ using $Eq.$ (7);
**10** $\quad\quad$ Calculate $L_{CBRD}$ as in $Eq.$ (8);
**11** $\quad\quad \theta \leftarrow \theta - \alpha \nabla_\theta L_{CBRD}$, with learning rate $\alpha$;
**12** $\quad$ Optimise the policy $\Phi_Q$ using $r_\theta(s, a)$ and $Eq.$ (5) ;

---

**Search Network Dataset :** The AMT dataset in Kant et al. (2022) contains 268 object categories in 12 different rooms and 32 receptacle types. Each object-room-receptacle (ORR) pair is ranked by 10 annotators in 3 classes: correct (positively ranked), misplaced (negatively ranked), and implausible (not ranked). For our problem statement, the misplaced class is of utmost importance. Hence, we rename the classes as (i) misplaced class $\rightarrow$ most probable class, (ii) correct class $\rightarrow$ less probable class, and (iii) implausible class remains the same. We find the ground truth score values for each ORR as the mean inverse of the ranks.

### 3.2 BENCHMARK DATASET FOR TESTING

The existing benchmark dataset, **RoomR** Weihs et al. (2021), has limitations as it only allows up to 5 objects, no object placement within another receptacle, and no blocked goal or swap cases. Thus, it cannot fully evaluate planning aspects such as the number of steps taken, agent traversal, blocked goal, or swap cases. To address this, we introduce **RoPOR**, a new benchmark dataset for testing task planners in Ai2Thor. It includes a diverse range of rooms (120) and object-receptacle pairs (118), allowing for a wide variety of rearrangement scenarios with up to 20 objects and random partial observability cases, object placement within receptacles in the current state, and blocked goal and swap cases. Moreover, object placement configurations in RoPOR affect sub-optimal planning policies in terms of agent traversal. The mean room dimensions along x-axis and y-axis are 3.12m and 5.80m, respectively. Refer Appendix for details on the distribution of objects, rooms and receptacles.

### 3.3 TRAINING

The training details of our Search network, Graph-based state Representation Network, Deep RL planner, and proxy reward network are available in the Appendix.

### 3.4 METRICS

Metrics in Weihs et al. (2021) do not highlight the efficacy of a task planner to judge efficient sequencing to reduce the number of steps taken or the agent traversal during rearrangement. For a fair evaluation of our method, and comparison against the existing methods and ablations, we define new metrics :

- **SNS** : **S**uccess measured by the inverse **N**umber of **S**teps uses a binary success rate ($S$) to evaluate the successful completion of a rearrangement episode along with the number of steps ($N_T$) taken by

| Number of Objects | Visible Objects | Unseen Objects | | Swap Case | Ours-GT | | | Ours | | | Weihs *et al.* | | | Gadre *et al.* | | | Sarch *et al.* | | | Ghosh *et al.* | | |
|---|---|---|---|---|---|---|---|---|---|---|---|---|---|---|---|---|---|---|---|---|---|---|
| | | OOF | OPR | | SNS↑ | ENR↑ | ATC(m) | SNS↑ | ENR↑ | ATC(m) | SNS↑ | ENR↑ | ATC(m) | SNS↑ | ENR↑ | ATC(m) | SNS↑ | ENR↑ | ATC(m) | SNS↑ | ENR↑ | ATC(m) |
| 5 | 5 | 0 | 0 | 0 | **0.98** | NC | **10.57** | 0.74 | NC | 11.98 | 0.018 | NC | 18.11 | 0.024 | NC | 20.15 | 0.058 | NC | 16.18 | 0.92 | NC | 13.58 |
| | 5 | 0 | 0 | 2 | **0.70** | NC | **12.36** | 0.53 | NC | 13.46 | 0 | NC | NC | 0 | NC | NC | 0 | NC | NC | 0.66 | NC | 16.73 |
| | 3 | 2 | 0 | 0 | **0.81** | **0.61** | **12.93** | 0.60 | 0.48 | 14.33 | 0.002 | 0.17 | 19.46 | 0.003 | 0.09 | 20.79 | 0.046 | 0.21 | 18.63 | 0 | NC | NC |
| | 3 | 0 | 2 | 0 | **0.79** | **0.60** | **13.39** | 0.58 | 0.47 | 14.89 | 0 | NC | NC | 0.0 | NC | NC | 0 | NC | NC | 0 | NC | NC |
| 10 | 10 | 0 | 0 | 0 | **0.97** | NC | **22.19** | 0.73 | NC | 24.51 | 0.002 | NC | 34.05 | 0.008 | NC | 36.69 | 0.032 | NC | 32.52 | 0.90 | NC | 27.98 |
| | 10 | 0 | 0 | 4 | **0.70** | NC | **24.63** | 0.52 | NC | 27.32 | 0 | NC | NC | 0 | NC | NC | 0 | NC | NC | 0.65 | NC | 30.45 |
| | 6 | 4 | 0 | 0 | **0.84** | **0.69** | **23.78** | 0.64 | 0.53 | 25.56 | 0.001 | 0.20 | 36.22 | 0.006 | 0.12 | 37.01 | 0.021 | 0.23 | 35.58 | 0 | NC | NC |
| | 6 | 0 | 4 | 0 | **0.83** | **0.67** | **24.15** | 0.62 | 0.52 | 25.97 | 0 | NC | NC | 0 | NC | NC | 0 | NC | NC | 0 | NC | NC |
| 20 | 20 | 0 | 0 | 0 | **0.95** | NC | **40.05** | 0.73 | NC | 44.05 | 0 | NC | NC | 0 | NC | NC | 0 | NC | NC | 0.88 | NC | 50.79 |
| | 20 | 0 | 0 | 8 | **0.70** | NC | **45.32** | 0.52 | NC | 48.32 | 0 | NC | NC | 0 | NC | NC | 0 | NC | NC | 0.62 | NC | 52.56 |
| | 12 | 8 | 0 | 0 | **0.87** | **0.75** | **41.29** | 0.67 | 0.58 | 45.29 | 0 | NC | NC | 0 | NC | NC | 0 | NC | NC | 0 | NC | NC |
| | 12 | 0 | 8 | 0 | **0.87** | **0.74** | **42.13** | 0.66 | 0.57 | 45.78 | 0 | NC | NC | 0 | NC | NC | 0 | NC | NC | 0 | NC | NC |

Table 1: (**OOF** : Objects outside agent's field of view initially, which are visible from a different perspective, **OPR** : Objects placed inside closed receptacles, **NC** : Not computable). When there are no unseen objects, the ENR is NC. Similarly, when SNS is zero, ENR and ATC are NC. Weihs *et al.*, Gadre *et al.*, and Sarch *et al.* do not handle 20 objects and cannot resolve swap cases without explicit buffer or OPR cases (**SNS = 0**). Ghosh *et al.* shows a slight decline in performance as the number of objects increase under complete visibility and swap cases, but fails to account for unseen objects. In comparison, **Ours** significantly outperforms Weihs *et al.*, Gadre *et al.* and Sarch *et al.* in terms of SNS, ENR, and ATC for visible objects, unseen objects, and swap cases without explicit buffer. Similarly, ours-GT performs better than Ghosh *et al.* in terms of **SNS** and **ATC** under complete visibility and swap cases without explicit buffer.

an agent to rearrange a given number of objects ($N$). $S$ is 1 if all object positions in the current and goal state are approximately equal. Higher the **SNS** implies a lower $N_T$ for a given $N$, indicating more efficient and successful rearrangement episode. (**SNS** = $S \times N/N_T$)

- **ENR**: **E**fficiency in **N**umber of **R**e-plans during object search by taking the ratio of the number of unseen objects initially ($N_{\widehat{V}}$) with respect to the number of attempts to search ($N_{S\widehat{V}}$). A higher **ENR** shows a lower $N_{S\widehat{V}}$ for a given $N_{\widehat{V}}$ indicating a more efficient search to find unseen objects. (**ENR** = $N_{\widehat{V}}/N_{S\widehat{V}}$)
- **Absolute Traversal Cost**($ATC$): The metric shows the overall distance traversed by the agent during the successful completion of a rearrangement episode. In an identical test configuration, a lower **ATC** indicates a more efficient rearrangement sequencing .

## 3.5 ABLATION

We ablate our task planner against ground-truth perception, various methods for object search and a dense reward structure. To study the effect of erroneous perception on our task planner, we assume the availability of **G**round-**T**ruth object detection labelling and 3D centroid localisation from Ai2Thor (*Ours-GT*). To understand the importance of our Search Network in planning, we replace it by a **(i) R**andom **S**earch policy (*Ours-RS*), which predicts probable receptacles for unseen objects with uniform probability and a **(ii) G**reedy **E**xploration strategy (*Ours-GE*) Chaplot et al. (2020) that optimizes for map coverage to discover all the unseen objects. To highlight the generalisation of proxy reward network to the overall objective of the rearrangement episode, we replace it with a hierarchical **D**ense **R**eward structure Ghosh et al. (2022) (*Ours-DR*) .Please refer to the appendix to find the results for the ablations, along with the analysis for the choice of hyper-parameters for each of our learning based modules.

## 3.6 QUANTITATIVE RESULTS

We evaluate our approach along with the existing methods on RoPOR - Benchmark Datset in Ai2Thor. Tab. 1 indicates that our method is scalable to large number of objects, as demonstrated by the consistent value of **SNS** despite the increasing number of objects across complete visibility, partial observability, and swap cases without an explicit buffer. The gradual increase in **ENR** with the increase in number of objects can be attributed to the fact that rearrangement of visible objects and the search for some unseen objects, indirectly aids in finding other unseen objects.

Comparing our method against Housekeep Kant et al. (2022) would be unfair because it does not perform a user-specific room-rearrangement with a pre-defined goal state. Instead, we have compared our method to previous works such as Weihs *et al.* Weihs et al. (2021), Gadre *et al.* Gadre et al. (2022), Sarch *et al.* Sarch et al. (2022) and Ghosh *et al.* Ghosh et al. (2022), all of which have demonstrated results for a user-specific room-rearrangement. For a fair comparison with Weihs *et al.*, we have used their best performing model - RN18+ANM, PPO+IL Weihs et al. (2021). Since, Ghosh *et al.*, uses groundtruth object positions in the current and the goal state, we compare it with our ablation method **Ours-GT**. Without erroneous perception, *Ours-GT* demonstrates efficient planning, by performing significantly better than all the existing methods Weihs et al. (2021); Gadre et al. (2022); Sarch et al. (2022); Ghosh et al. (2022), including *Ours*, in terms of **SNR**, **ENR** and **ATC**.

**Under complete visibility**, *ours* significantly outperforms Weihs *et al.*, Gadre *et al.* and Sarch *et al.* in terms of **SNS** and **ATC**. Similarly, *Ours-GT* significantly outperforms Ghosh *et al.* in terms of **ATC**. The improvement over Weihs *et al.*, Gadre *et al.* and Sarch *et al.* shows their heuristic planner is neither scalable nor does it optimize the overall agent traversal or the number of rearrangement steps. In contrast, our method leverages compact graph-based scene geometry capable of addressing large numbers of objects, and robust Deep RL makes our planner efficient in reducing the redundant traversal of the agent. Our method uses path length cost and proxy reward with the episodic notion, which helps to improve the overall traversal of the agent to produce lower **ATC**. In comparison, Ghosh *et al.* uses greedy Euclidean distance based reward without having an episodic notion, thus failing to optimize overall traversal. Moreover, Ghosh *et al.* shows a drop in performance on the RoPOR dataset as compared to their results evaluated on RoomR Weihs et al. (2021), due to the variations in the testing scenarios in RoPOR that significantly impact agent traversal for sub-optimal rearrangement policies.

**Under partial observability**, there are two cases - **(i) OOF**: Objects located outside the field of view initially which are visible from a different perspective and **(ii) OPR**: Objects placed inside closed receptacles. In the case of **OOF**, our method substantially outperforms Weihs *et al.*, Gadre *et al.* and Sarch *et al.* in terms of **SNS**, **ENR** and **ATC**. All these above methods use greedy sub-optimal planners and employ explicit scene exploration to find objects outside the field of view, incurring huge traversal cost as indicated by their **ATC**. To gauge the performance of the exploration strategy for object search in terms of **ENR**, we consider each newly generated location or a set of navigational steps from the exploration policy as a search attempt. Our approach's significantly higher **ENR** shows that the Search Network outperforms the exploration policies of Weihs et al. (2021); Gadre et al. (2022); Sarch et al. (2022) in terms of the number of attempts to find unseen objects. Ghosh *et al.* does not address any case of partial observability. While Weihs *et al.*, Gadre *et al.* and Sarch *et al.* do not solve the case of **OPR**, which involves object placement inside receptacles (**SNS = 0**). However, our approach performs equally well in both cases of partial observability due to our search network's ability to comprehend a commonsense based semantic relationship between an object and any type of receptacle - rigid or articulated.

**Swap cases** without an explicit buffer are not handled by Weihs *et al.*, Gadre *et al.* and Sarch *et al.*, which is evident from **SNS = 0**. *Ours*, *Ours-GT* and Ghosh *et al.* can effectively resolve an increasing number of swap cases without an explicit buffer using the hybrid action space Ghosh et al. (2022) in the Deep RL network. However, *Ours-GT* performs better than Ghosh *et al.* in terms of **ATC** due to a novel *collsion resolution reward* that optimizes the agent's traversal.

To ground the values of our RoPOR dataset, we show the results for Ours, the ablation methods and the SOTA in the test set of RoomR in the Appendix. Moreover, additional results for individual methods in our pipeline can be found in the Appendix.

### 3.7 Qualitative Results

To show the results of our method in room-rearrangement, we have created videos in a number of test scenarios to highlight the robustness of our method. We also test our method in a new environment - Habitat, as demonstrated in our supplementary video. This transfer does not require any additional training for our Search Network, Graph-based State Representation or Deep RL planner. This shows the capability of our method for seamless sim-to-sim transfer, further emphasizing its suitability for real-world deployment. Please refer the supplementary video.

## 4 Limitations

Our approach is not capable of identifying unseen objects that are occluded due to clutter on receptacles (for e.g. a spoon may become occluded, if bread, box, lettuce etc. is placed before it). Our method also assumes the availability of perfect motion planning and manipulation capabilities.

## 5 Conclusion

This paper presents an innovative task planner designed for organizing rooms under conditions of partial observability. Our approach minimizes agent traversal and step count during both object search and rearrangement by leveraging a Search Network followed by a Deep RL-based planner. By utilizing a graph-based state representation and episodic proxy reward, our method exhibits versatility and applicability across a range of scenarios. The RoPOR benchmark dataset facilitates additional research in the realm of Embodied AI-based rearrangement. Future endeavors will concentrate on deploying our approach in real-world settings.

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

## A  APPENDIX

This supplementary document first shows our results on **RoomR** Weihs et al. (2021) dataset. Subsequently, we discuss our RoPOR dataset - statistics and configurations. Detailed information about our reward structure and rearrangement algorithm. Additionally, we provide our network details and training schedule. Finally, we show our ablation study and discuss our failure scenarios.

## B  RESULTS ON ROOMR DATASET (WEIHS ET AL. (2021))

Tab. 2 shows the results for *Ours*, the ablation methods and the SOTA in the test set of RoomR. The similarity in the results of our ablation methods in RoomR reflects the redundancy in the dataset and its metrics to gauge the planning efficacy. Both Ghosh et al. (2022) and *Ours-GT* use ground-truth perception, hence comparing them with other methods would be unfair. *Ours* outperforms all the existing methods in RoomR, showing the generalizability and robustness of our method.

## C  ROPOR BENCHMARK DATASET

The existing datasets and benchmarks for room rearrangement, such as RoomR Weihs et al. (2021), contain at most five objects in a scene. Further, this dataset and benchmark does not explicitly mention the inclusion of different configurations required to test the efficacy of rearrangement sequencing. To this end, we propose our benchmark dataset RoPOR, which contains various configurations for swap and object placements. Moreover, to cater to the needs of object rearrangement under partial observability, we take inspiration from Housekeep Kant et al. (2022) to define the distribution of objects, rooms, and receptacles based on the ground-truth human annotations from AMT Kant et al. (2022).

### C.1  DISTRIBUTION OF OBJECT-ROOM-RECEPTACLES

To create realistic untidy scenarios in the RoPOR dataset, object placement on the room-receptacles is not entirely random but follows certain semantic prior to reflect the underlying human preferences for untidiness. This semantic prior is induced using approximately 34,000 annotations, where each annotation comprises of rankings from 10 individuals, for each pair of 128 room-receptacles and 269 objects in the AMT human preference dataset (Kant et al., 2022). A plausible room-receptacle for a given object is randomly selected based on the annotations. In each room of the RoPOR dataset, we ensure to keep a higher number of objects with a misplaced class of room-receptacles than with a correct or implausible class of room-receptacles. In Fig. 4, we show the average percentage of objects placed in each room-receptacle class for any given room with a fixed number of receptacles in the RoPOR dataset. As the room is untidy, there is a greater likelihood of finding objects in misplaced

| | *Ours* | *Ours-RS* | *Ours-GE* | *Ours-DR* | Weihs et al. (2021) | Gadre et al. (2022) | Sarch et al. (2022) | *Ours-GT* | Ghosh et al. (2022) |
|---|---|---|---|---|---|---|---|---|---|
| Success Rate ↑ | **0.43** | 0.42 | 0.42 | 0.40 | 0.003 | 0.004 | 0.024 | 0.98 | 0.97 |
| Fixed Strict ↑ | **0.519** | 0.48 | 0.46 | 0.41 | 0.014 | 0.019 | 0.116 | 0.99 | 0.98 |
| Energy Remaining ↓ | **0.631** | 0.66 | 0.68 | 0.71 | 1.10 | 1.17 | 0.931 | 0.023 | 0.031 |

Table 2: Result of our method, the ablations and the existing methods on the Test set of RoomR and its metrics.

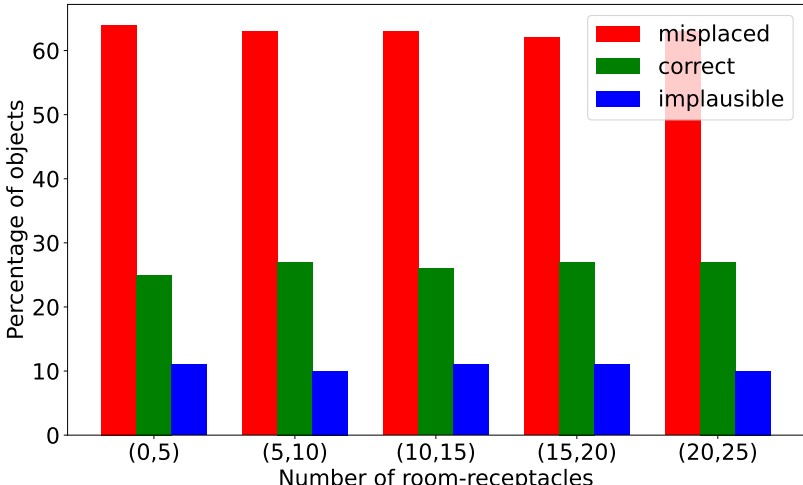

Figure 4: Distribution of objects in a room of RoPOR dataset for each class of room-receptacle, i.e., misplaced, correct, and implausible as per AMT dataset.

room-receptacles compared to finding them in their correct room-receptacles. Additionally, we place a small percentage of objects in the implausible class of room-receptacles to simulate complete disorderliness.

Moreover, it is logical to have things in more disordered configurations for an untidy room, hence a higher representation of object-room receptacles from the misplaced and implausible class. The graph in Fig. 5 shows the distribution of the high-level object categories in the misplaced and the correct class of room-receptacles for the RoPOR dataset. The high-level categories in the dataset are shown in Tab. 8. It is evident from Fig. 5 that the number of misplaced room-receptacle is greater than the number of correct room-receptacles for most of the high-level object categories in the RoPOR dataset. As we have borrowed the ground truth classes from AMT, our results show a similar distribution to AMT in Housekeep.

## C.2 OBJECT CONFIGURATIONS

### C.2.1 INCLUSION OF SUFFICIENT SWAP CASES

In every testing scenario, we have two types of swap configuration : (i) With a distance of more than 1m, in which minimising the traversal distance is more important than minimising the number of pick-place actions and (ii) With a distance of less than 1m, in which minimising the traversal distance is insignificant compared to minimising the number of pick-place actions. This testing scenario is depicted in Fig. 9.

### C.2.2 IMPORTANCE OF PATH LENGTH DISTANCE

To emphasize the importance of path length distance during rearrangement, we ensure to have sufficient representation of object configurations in each room so that the objects are closer in terms of euclidean distance than the path length distance and vice versa. An example of this configuration can be seen in Fig. 7, where lettuce (violet) is farther from the goal position (blue) of tomato (red) than the kettle (pink) in terms of path length distance due to the table. Also, this ensures that the lettuce is closer than the kettle to the goal position of the tomato in terms of Euclidean distance.

### C.2.3 IMPORTANCE OF NON-GREEDY POLICY

In order to account for greedy policies during rearrangement, we have included scenarios in the RoPOR dataset which lead to poor performance in terms of agent traversal for policies that choose only the nearest object. For instance, see Fig. 6, where the policy which chooses the immediate nearest object after every move leads to a longer traversal path for the agent. After a similar step 1, the greedy policy chooses to pick-place lettuce (violet) which is closest in terms of the Euclidean distance. In contrast, an optimal policy chooses the kettle (pink), which is farther in terms of Euclidean distance.

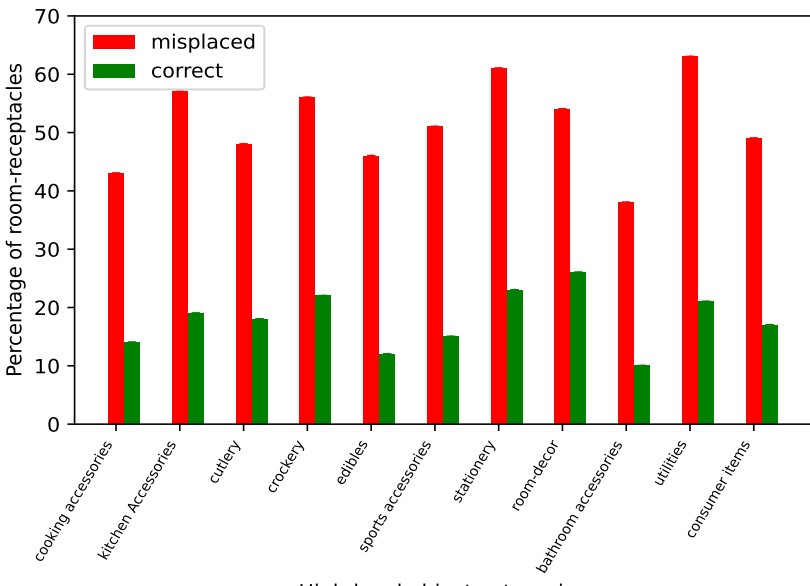

Figure 5: Distribution of room-receptacles for the high-level object categories (Tab. 8) in the RoPOR dataset. This graph shows the percentage of room-receptacles of the misplaced and correct class (as per AMTKant et al. (2022)) in RoPOR.

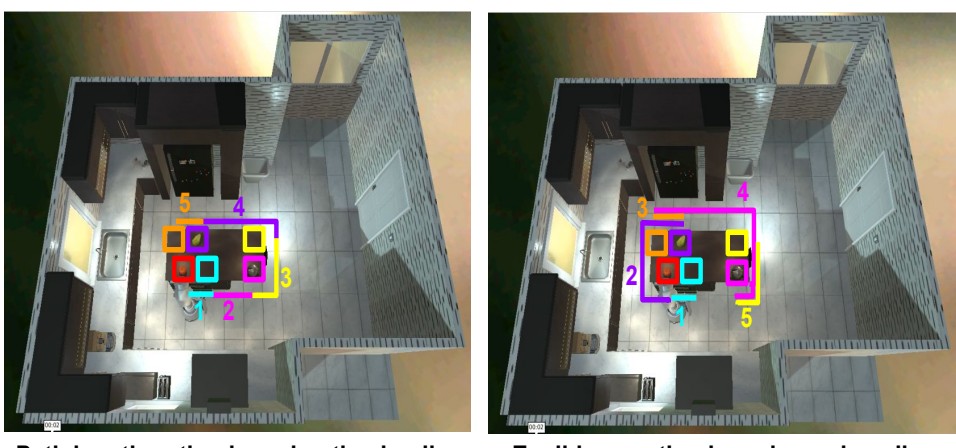

**Path length notion based optimal policy**     **Euclidean notion based greedy policy**

Figure 6: A sample run of a Euclidean distance-based greedy policy and a Path length distance-based optimal policy on the configuration shown in Fig. 7. Both policies traverse the same step 1 (shown in sky blue) to pick and place the red object. For step 2, the greedy policy based on euclidean distance chooses the violet object as it is closer (in terms of Euclidean distance). However, the path length distance-based optimal policy chooses the pink object (as it is closer in terms of the path length distance and more optimal considering the overall rearrangement scenario). Hence, the path traversed in step 2 (shown in violet for greedy euclidean and pink for optimal-path length) is much greater for Euclidean-greedy policy than the optimal-path length one. Moreover, due to repeated greedy actions without the notion of an overall rearrangement scenario, the greedy policy traverses a much longer path than the optimal one.

The optimal policy-based method chooses a step taking into account the whole rearrangement episode, which leads to a shorter traversal path overall. After taking greedy step 2, the greedy policy leads to a much longer path traversed.

Finally, we have shown a sample configuration for 5 and 10 objects, including all the scenarios mentioned above in Fig. 8.

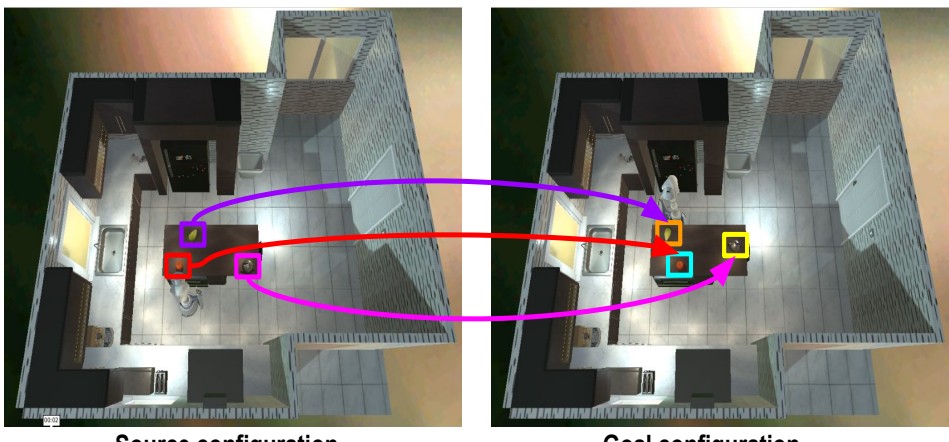

**Source configuration**          **Goal configuration**

Figure 7: An example configuration of a source and goal with three objects in RoPOR dataset signifying the importance of path length distance over euclidean distance. The source location for the object in the violet box is closer to the goal position (blue box) for the object in the red box in terms of the Euclidean distance. In contrast, the object in the pink box is closer to the blue box in terms of the path length distance.

## D   REWARD STRUCTURE

We use a dense reward structure to compute the per-step reward and store it in the replay buffer. The computed rewards train our proxy network, and the step-wise proxy reward from the proxy reward network is used to train our Deep RL method. The proxy reward imparts episodic awareness and improves the sampling efficiency during our Deep RL training. Inspired by the work in Ghosh et al. (2022), we use a similar reward structure so that our Deep RL method minimizes the number of moves and reduces the overall traversal of the agent. We take special care for swap instances to strike a balance between (i) optimizing the number of moves and (ii) minimizing the overall traversal, unlike Ghosh *et al.*, which focuses only on (i). Further, we describe in detail the dense reward structure in Fig. 10, which enables our RL to produce an effective plan to move the correct objects and efficiently handle the blocked goals, specifically swap instances.

- **Infeasible action reward** $R1$**:** The objective of this reward is to restrain the agent from producing infeasible actions that can not be realized in the environment. In Fig. 10, $R1$ block shows this reward structure.
- **Collision resolution reward** $R2$ **:** This reward tries to strike a balance between (i) optimizing the number of moves and (ii) reducing the overall traversal of the agent. Fig. 9 shows an example scenario where objects A (pink) and B (blue)in the right image are far apart (more than 1m in the path length distance), whereas object D (pink) and C (blue) in the left image are close to each other. This reward ensures that the traversal is optimized for the case of A and B by slightly compromising the number of moves (one extra pick-place step). In contrast, in the case of D and E, we slightly comprise on the traversal cost to optimize the number of moves. It prioritizes the goal-occupied objects (objects which occupy the goal location of other objects) to move first instead of the goal-blocked objects. For example, moving B and C first helps free the space for the goal-blocked objects (A, D) before they move to resolve the collision. It also ensures that the free location of goal occupied object will be nearest to the goal-occupied object's target location instead of placing it in any random location. Instead of moving A to B's source location, this reward moves A to its nearby location and frees up B's goal location. This enforces that the traversal of the agent is minimized. In Fig. 10, $R2$ block describes this reward.
- **Nearest neighbour reward** $R3$ **:** We use this reward (defined in $R3$ block of Fig. 10) to ensure that the agent's traversal should be minimal by first arranging the nearest objects from the previously placed object rather than arranging the objects randomly.

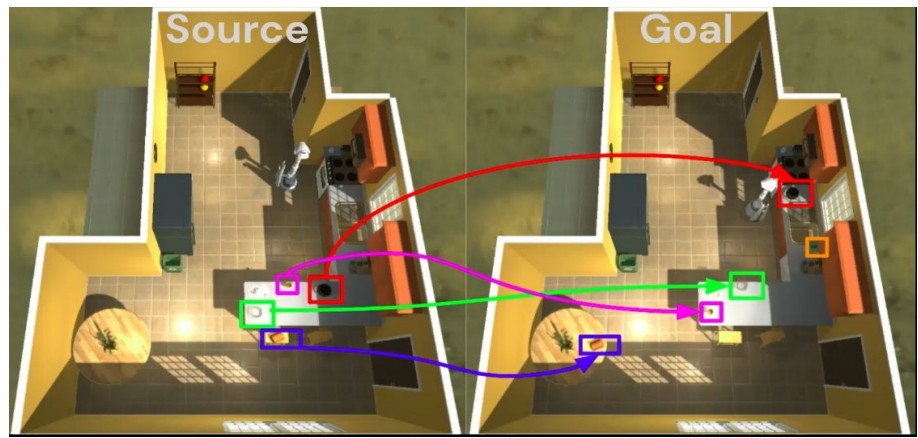

**5 objects room - rearrangement**

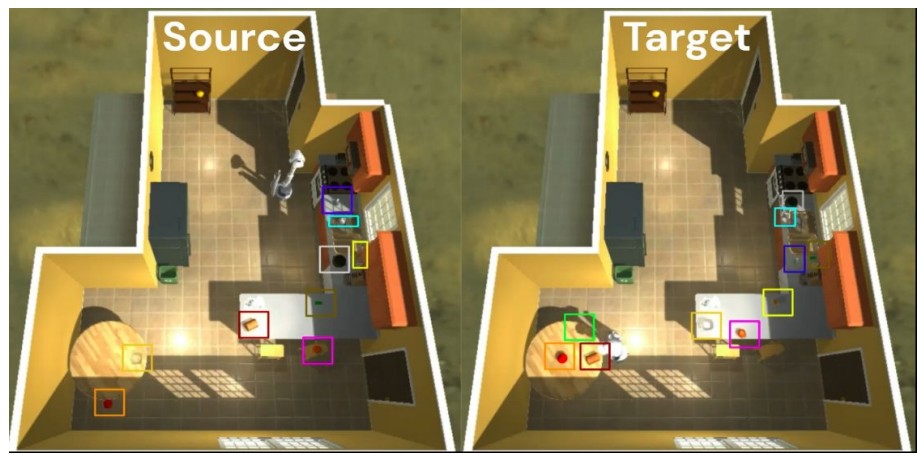

**10 objects room - rearrangement**

Figure 8: Rearrangement configuration under partial observability and swap cases for 5 and 10 objects. This is a sample configuration of objects in our dataset, with various configurations for swap, path length distance, and non-greedy policies.

- **Goal reaching reward for object and receptacle** $R4$**:** We use this reward (depicted in $R4$ block of Fig. 10) to eliminate erroneous redundant actions. This reward ensures that our method can place the object in its goal location in a single move whose goal position is free. It penalizes negative residual Euclidean distance if it fails to place the object in its goal location.
- **Episodic Traversal Cost :** We use a novel episodic reward to impart the notion of path length traversed along with the number of steps taken in an episode as shown in Eq. (6).

## E  ALGORITHM FOR OUR METHOD

We present the pseudo-code of our method in Algorithm 2. From the goal state room $\mathcal{G}$, we capture the object list $\mathbf{O} = \{[\mathbf{W}_i, \mathbf{P}_i]\}_{i=1,2,..,N}$ for all the detected objects as shown in Line 1. Moreover, we also get the room-receptacle list $\mathbf{R} = \{[\mathbf{W}_i^R, \mathbf{P}_i^R], i = 1, 2, .., N_R\}$. Also, we generate the 2D occupancy grid map $\mathbf{M}^{2D}$ and 3D map $\mathbf{M}^{3D}$ from the room in goal state. Using this information, we create a goal state graph ($G_g$) using $O$ as described in Line 2 and Line 3. From the image $I_t$ in the current state at time $t$, we get the semantic labels and positions of the visible objects as $\mathbf{O}^V = \{[\mathbf{W}_i^V, \mathbf{P}_i^V], i = 1, 2, .., N_V\}$. Comparing $\mathbf{O}$ in the goal state with $\mathbf{O}^V$ in the current state allows for determining only the semantics of unseen objects $\mathbf{O}^{\widehat{V}} = \{\mathbf{W}_i^{\widehat{V}}, i = 1, 2, .., N_{\widehat{V}}\}$. Our goal is to solve the rearrangement sequence for the visible objects ($O^V$) and search the unseen ones ($\mathbf{O}^{\widehat{V}}$). The Search network takes the input as the pairwise concatenated names of $W^{\widehat{V}}$ and $W^R$ to generate

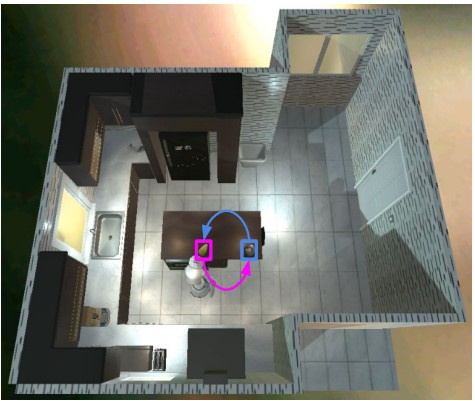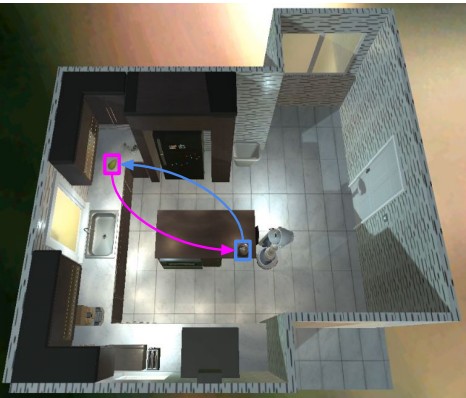

Figure 9: An example of two types of swap cases. The left image shows the swap between the pink and blue objects at a shortest path length distance of $0.55m$, whereas the right image shows a swap instance with an object separation of $2.5m$ in terms of the shortest path length distance. For the swap instance in the left image, if we reduce the traversal to $2 \times 0.55m$, we end up taking $1$ more pick-place move, whereas if we compromise on the traversal, we end up with $1$ less move and traverse only $0.55m$ more. For the swap instance on the right, if we reduce the number of pick-place moves by $1$, we traverse $2.5m$ more, whereas if we compromise on the number of moves, we complete the swap by just traversing $2 \times 2.5m$.

the position of probable receptacles ($P^{\widehat{V}R}$). We generate the current state graph as shown in Line 10 and Line 11, using $O^V$ for visible objects and, $O^{\widehat{V}}$ and $P^{\widehat{V}R}$ for the unseen ones. The Graph Siamese Encoder Network(GSEN) is used to calculate the embedding $Z_p$ from the graph representation $G_c$ and $G_g$. We find the binary collision vector ($C$) by computing the similarity between $P^V$ and $P$. Using the state space $s$, as computed in Line 14, in the Deep RL network, we generate a sequence of actions(a) such that $a = \{a_i = (k_i, p_{k_i})\}_{i=1,2,..,K}$. This rearrangement continues unless the $a$ becomes empty ($\phi$) or the maximum number of steps are exhausted ($T_M$).

Each action in the sequence $a$ consists of individual action $a_i = (k_i, p_{k_i})$, where $k_i$ is the discrete action which specifies the objects that needs to be moved or searched and $p_{k_i}$ is the continuous parameter to specify the location of the pick-place or the search. If $k_i$ points towards an unseen object, say $o^{\widehat{V}} \in O^{\widehat{V}}$ as shown in Line 21, then a search action is performed on the most probable receptacle whose position is $p^r \in P^R$ and name is $w^r \in W^R$. The agent looks for unseen object $o^{\widehat{V}}$ at $p^r$ and along the path from the agent's current position to $p^r$, unless the maximum number of attempts are exhausted, as shown in Line 24. In case the predicted receptacle is articulated, the agent opens it and looks for the object. After the search loop ends, we remove the searched receptacle ($w^r$) from $W^R$ as it no longer contains any information about other unseen objects which is depicted in Line 29. If $k_i$ points towards a visible object whose position $P_i$ in the goal state is known, then we perform a pick-place action to place $k_i$ as close as possible to its target $P_i$ as in Line 32.

### E.1 RE-PLANNING STRATEGY

The agent needs to re-plan a new sequence of actions if, **(i)** the agent fails to find the unseen object at the predicted receptacle as in Line 29. In this case, the agent discards the predicted receptacle ($W^R \setminus w^r$) from further search attempts and also predicts a new receptacle for the unseen object from the Search Network. **(ii)** The agent finds the unseen object, it is searching for, before arriving at the predicted receptacle as shown in Line 27, then the agent updates $O^V$ and $O^{\widehat{V}}$. These updates are reflected in $G_c$ and using the state representation embeddings from the graph, the Deep RL understands the objects which need to be searched for or rearranged and accordingly generates a new sequence.

## F EXPLORATION AND PATH PLANNING

Our method uses a similar exploration strategy as in Sarch *et al.*Sarch et al. (2022). We explore the scene using a classical mapping method. We take the initial pose of the agent to be the fixed coordinate frame in the map. We rotate the agent in-place and use the observations to instantiate

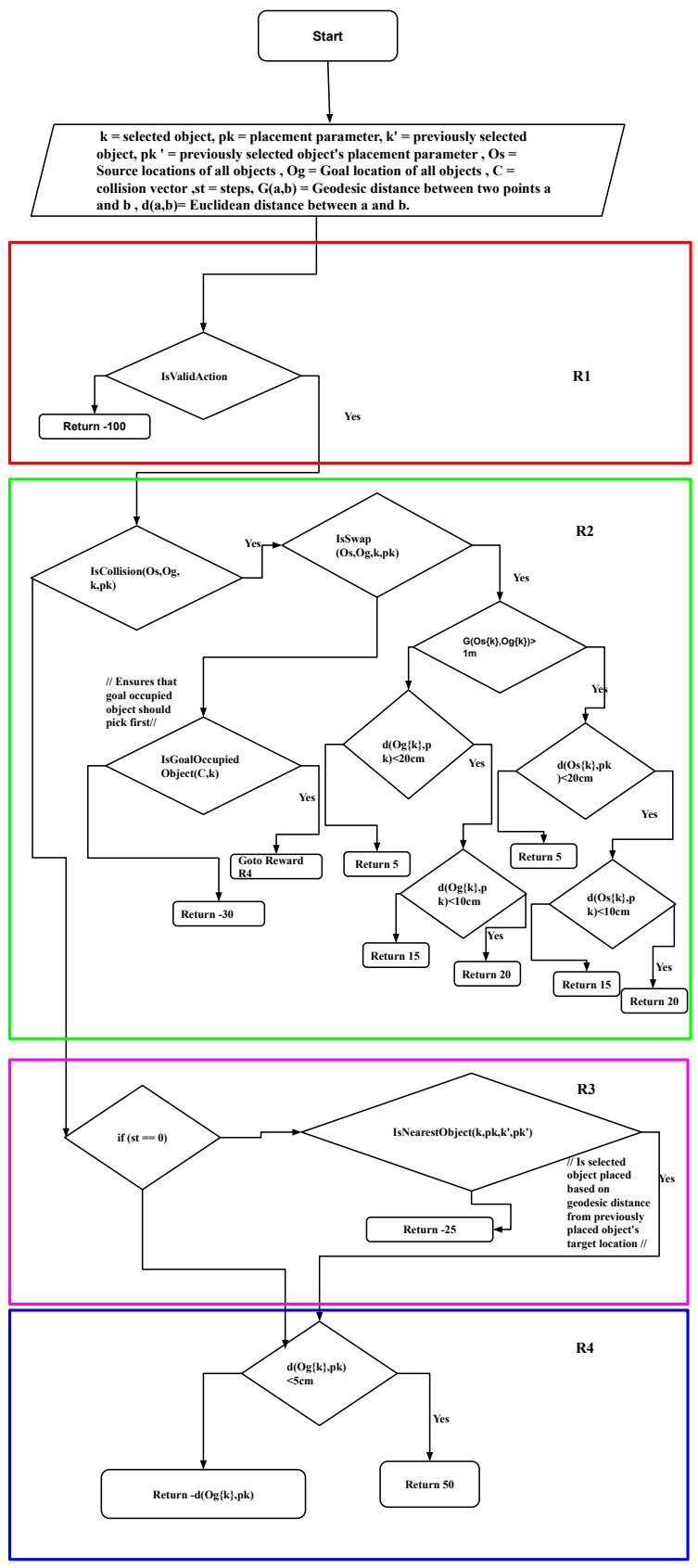

Figure 10: Algorithm of our hierarchical dense reward structure.

**Algorithm 2:** Task planner

---

**Input:** $I_t, T_M$
**Data:** $RobertaModel, RobertaTokenizer$
**Result:** Sequence of actions $a = \{a_0, .., a_K\}$

1   $O = \{[\mathbf{W}_i, \mathbf{P}_i]\}_{i=1}^N, R = \{[\mathbf{W}_i^R, \mathbf{P}_i^R]\}_{i=1}^{N_R}, M^{\text{2D}}, M^{\text{3D}} \leftarrow \mathcal{G}$;

2   $V(G_g) \leftarrow \{O\}$;

3   $E(G_g) \leftarrow \mathcal{D}(P_i, P_j)_{i \neq j}$ on $M^{\text{2D}}$;

4   $t_1 \leftarrow 0$;

5   **while** $t_1 \leq T_M$ **and** $(W \setminus W^V \neq \phi$ **or** $P^V \neq P)$ **do**

6      $O^V = \{[\mathbf{W}_i^V, \mathbf{P}_i^V]\}_{i=1}^{N_V} \leftarrow I_t$;

7      $O^{\widehat{V}} \leftarrow O \setminus O^V$;

8      $P^{\widehat{V}R} \leftarrow SearchNetwork(W^{\widehat{V}} \times W^R)$;

9      $\overline{P} \leftarrow P^V \cup P^{\widehat{V}R}$;

10      $Nodes(G_c) \leftarrow \{O^V, O^{\widehat{V}} \cup P^{\widehat{V}R}\}$;

11      $Edges(G_c) \leftarrow \mathcal{D}(\overline{P}_i, \overline{P}_j)_{i \neq j}$ on $\mathbf{M}^{\text{2D}}$;

12      $Z_p \leftarrow GSEN(G_c) \cup GSEN(G_g)$;

13      $C \leftarrow CollisionDetection(P^V \cap P)$;

14      $s \leftarrow Z_p \cup C$;

15      $a = \{a_i = (k_i, p_{k_i})\}_{i=1,..K} \leftarrow DeepRL(s)$;

16      $\overline{O}^V \leftarrow O^V$;

17      $\overline{W}^R \leftarrow W^R$;

18      $t_2 \leftarrow 0$;

19      **while** $t_2 \leq T_M \& (a \neq \phi \& \overline{O}^V == O^V \& \overline{W}^R == W^R)$ **do**

20          $(k_i, p_{k_i}) \leftarrow a_i$;

21          **if** $k_i \in O^{\widehat{V}}$ **then**

22              $\{w^r, o^{\widehat{V}}, p^r\} \leftarrow k_i$;

23              $t_3 \leftarrow 0$;

24              **while** $t_3 \leq T_M \& ($**not found** $o^{\widehat{V}}$ **in Search** $(w^r, p^r))$ **do**

25                  $Update\{O^V, O^{\widehat{V}}\} \leftarrow I_{t+1}$;

26                  **if** $o^{\widehat{V}} \in O^V$ **then**

27                      break;

28                  $t_3 \leftarrow t_3 + 1$;

29              $W^R \leftarrow W^R \setminus w^r$;

30          **else**

31              $t_4 \leftarrow 0$;

32              **while** $t_4 \leq T_M \& PickPlace(k_i) \neq P_i$ **do**

33                  $Update\{O^V, O^{\widehat{V}}\} \leftarrow I_{t+1}$;

34                  $t_4 \leftarrow t_4 + 1$;

35          $a_i \leftarrow a_{i+1}$;

36          $t_2 \leftarrow t_2 + 1$;

37      $t_1 \leftarrow t_1 + 1$;

an initial map. Second, the agent incrementally completes the maps by randomly sampling an unexplored, traversable location based on the 2D occupancy map built so far and then navigates to the sampled location, accumulating the new information into the maps at each time step. The number of observations collected at each point in the 2D occupancy map is thresholded to determine whether a given map location is explored or not. Unexplored positions are sampled until the environment has been fully explored, meaning that the number of unexplored points is fewer than a predefined threshold.

To navigate to a goal location, we compute the shortest path length to the goal from the current position using the Djikstra algorithm, given the 2D occupancy map $\mathbf{M}^{\text{2D}}$.

## G  NETWORKS

### G.1  SEARCH NETWORK

- Generate RoBERTa embedding : Each object and room-receptacle label is concatenated together and encoded into an embedding($E_{\widehat{V}_R}$) of size 1024 using the average of the output from the second last hidden layer of pre-trained RoBERTa large model.
- Sorting Network(SRTN): The combined embedding($E^{\widehat{V}R}$) of the object-room-receptacle is passed through SRTN consisting of 4 fully connected(FC) layers followed by ReLU and softmax, which outputs the softmax probability of an object being at that room receptacle represented by the three classes: 1) Most probable class, 2) Less probable class and 3) Implausible class. These FC layers contain a dropout of 0.2, i.e. $FC(E_{\widehat{V}_R}, 512) \rightarrow ReLU \rightarrow FC(512, 256) \rightarrow ReLU \rightarrow FC(256, 64) \rightarrow ReLU \rightarrow FC(64, 3) \rightarrow softmax$.
- Scoring Network(SCN): The embeddings of the most probable class($E^0_{\widehat{V}_R}$) and the less probable class($E^1_{\widehat{V}_R}$) is passed through the SCN consisting of 3 fully connected(FC) layers followed by ReLU, which outputs a score($\widehat{\chi}$) denoting the probability of finding an object at that room-receptacle. These FC layers contain a dropout of 0.2, i.e. $FC(E^{0,1}_{\widehat{V}_R}, 256) \rightarrow ReLU \rightarrow FC(256, 64) \rightarrow ReLU \rightarrow FC(64, 1)$. The $argmax$ of the probabilities decides the class to which the combined embedding belongs. For the objects in the implausible class the $\widehat{\chi} = 0$ is used.

### G.2  GRN

- **GSEN :** It takes $G_c$ and $G_g$ as input and uses a Siamese network to encode $G_c$ and $G_g$ . Each encoder of the Siamese network consists of 2 fully connected GraphConv followed by RELU. i.e. $GraphConv(G_c, 256) \rightarrow RELU \rightarrow GraphConv(256, 128)$ and $GraphConv(G_g, 256) \rightarrow RELU \rightarrow GraphConv(256, 128)$.
- **RGDN :** It takes the combined embedding of $G_c$ and $G_g$ as $Z_p$ which is passed through 2 fully connected (FC) layers followed by RELU and produces $\tau_p$. These FC layers contain a dropout of 0.25. i.e. $FC(Z_p, 64) \rightarrow RELU \rightarrow Dropout(0.25) \rightarrow FC(64, \tau_p)$.

### G.3  PROXY REWARD NETWORK

It takes input as $Z_p \cup C \cup a \cup \bar{Z}_p \cup \bar{C}$, where $\bar{Z}_p \cup \bar{C}$ is the graph embedding and collision vector to represent the state space of the next state. The network consists of 4 fully connected networks followed by RELU. These FC layers contain a dropout of 0.25, i.e. $FC(Z_p \cup C \cup a \cup \bar{Z}_p \cup \bar{C}, 512) \rightarrow RELU \rightarrow Dropout(0.25) \rightarrow FC(512, 128) \rightarrow RELU \rightarrow Dropout(0.25) \rightarrow FC(128, 64) \rightarrow RELU \rightarrow Dropout(0.25) \rightarrow FC(64, 1)$.

### G.4  DEEP RL

- ($\Phi_Q$) **:** It takes input state as $Z_p \cup C$ and all action parameters $p$, and uses 3 fully connected (FC) layers followed by RELU, which outputs discrete actions K. These FC layers contain a dropout of 0.5, i.e. $FC(Z_P \cup C \cup p, 512) \rightarrow RELU \rightarrow Dropout(0.5) \rightarrow FC(512, 128) \rightarrow RELU \rightarrow Dropout(0.25) \rightarrow FC(128, K)$.
- ($\Phi_P$)**:** It takes input as $Z_p \cup C$, and uses 3 fully connected (FC) layers followed by RELU, which outputs all action parameter p. These FC layers contain a dropout of 0.5, i.e. $FC(Z_p \cup C, 512) \rightarrow RELU \rightarrow Dropout(0.5) \rightarrow FC(512, 128) \rightarrow RELU \rightarrow Dropout(0.25) \rightarrow FC(128, p)$.

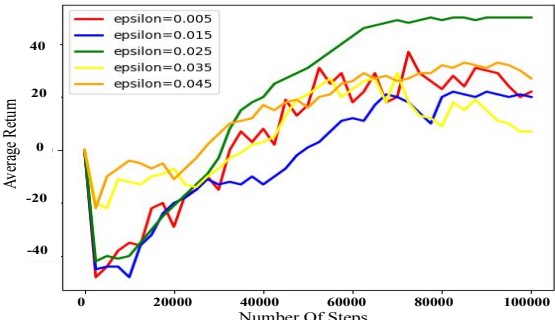

Figure 11: Comparison of the effect of $\epsilon$ on average return with increasing steps. $\epsilon = 0.025$ gives the best average return.

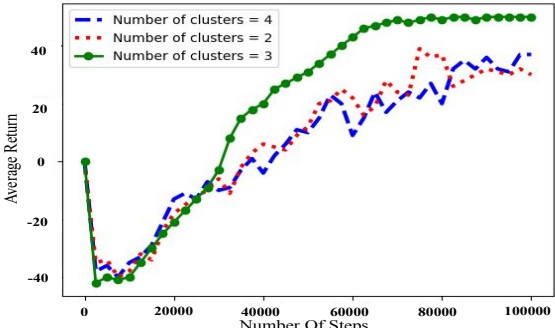

Figure 12: Comparison between the different number of clusters for the training of proxy reward using CB-RD. Number of clusters = 3 gives the best average return indicating a better sampling efficiency.

## H  TRAINING METHOD

### H.1  SEARCH NETWORK

For training the Search network, we use the Search Network dataset, which contains the modified classes of the AMT dataset with the ground truth class labels and the probability score calculated from the mean inverse of Human annotated ranks. The entire dataset is split into train, val, and test sets with a ratio of $55 : 15 : 30$.

- Training SRTN: To train SRTN, we use an Adam optimizer with a learning rate($\alpha_{SR} = 0.001$) and a weight decay rate($\lambda_{SR} = 0.0001$). Further, we use a randomized weighted data sampler to equalize the classes in each batch size of 512. The $argmax$ of softmax probabilities for each ORR in the three classes decides the class to which the combined embedding belongs. This is used to calculate the cross entropy loss(Eq.(1)) using the ground truth class labels in the Search Network dataset. This loss is backpropagated during each epoch to train the network.
- Training SCN: To train SCN, we use the ORR belonging to the Most probable and the Less probable class in the Search Network dataset. Further, we use an Adam optimizer with a learning rate($\alpha_{SC} = 0.001$) and a weight decay rate($\lambda_{SC} = 0.01$). Further, we use a randomized weighted data sampler to equalise the classes in each batch size of 512. The output is used to calculate the mean square error loss(Eq.(2)) using the ground truth probability score in the Search Network dataset. This loss is backpropagated during each epoch to train the network.

### H.2  GRN

The two encoders of GSEN use share weights to produce embeddings $Z_c$ and $Z_g$ for $G_C$ and $G_g$ respectively . To train our GRN, we use an Adam optimizer with a learning rate($\alpha_{GRN} = 0.01$).

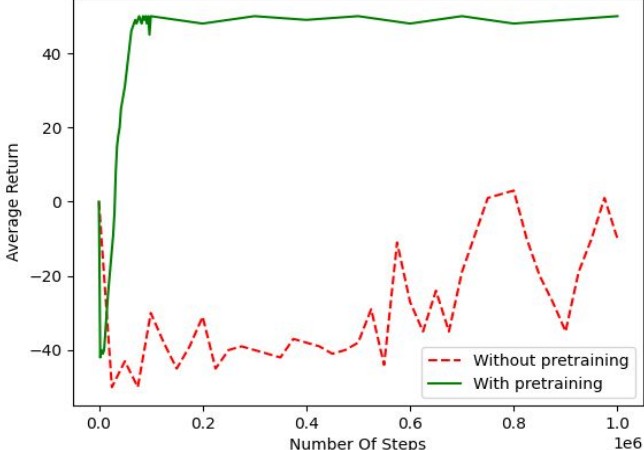

Figure 13: The plot demonstrates the effectiveness of a pre-trained GRN in encoding relative distance notion well.

Further, we use a randomized data sampler with a batch size of 512. The output is used to calculate the mean square error loss (Eq.(3)) using the ground truth $\tau$.

### H.3 DEEP RL AND PROXY

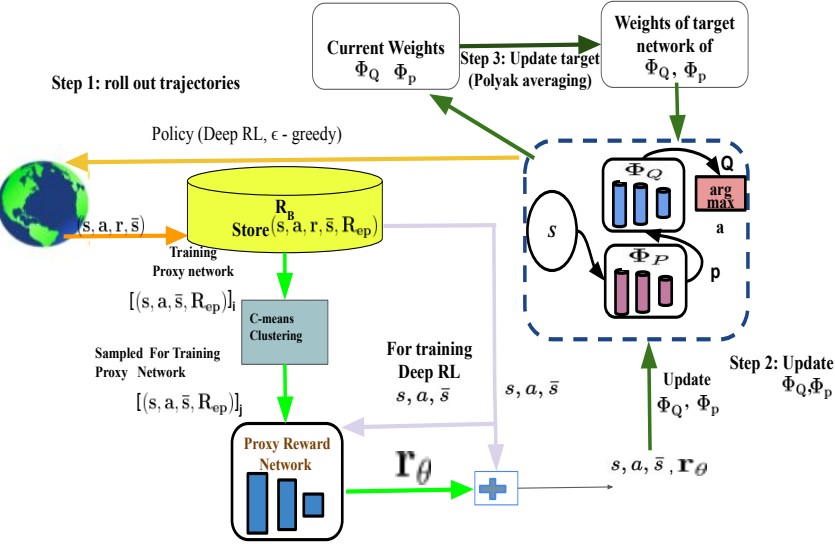

Figure 14: The diagram shows the training procedures of our Proxy Network and Deep RL method. We train the Deep RL in a 3-step method by fetching the samples from the replay buffer and using the predicted reward from the Proxy Network. We train the Proxy Network using the reward coming from the replay buffer.

Fig. 14 shows our overall training method to train our Deep RL and Proxy Network. We train the Proxy Network using the loss shown in Eq. (8). We use off-policy method to train our RL. Our training method consists of the following three steps

- **Step 1:** We use $\epsilon$-greedy exploration to generate trajectories $\{s, a, r, \bar{s}\}$ and store into replay buffer. Here the size of the replay buffer is 1000000000.

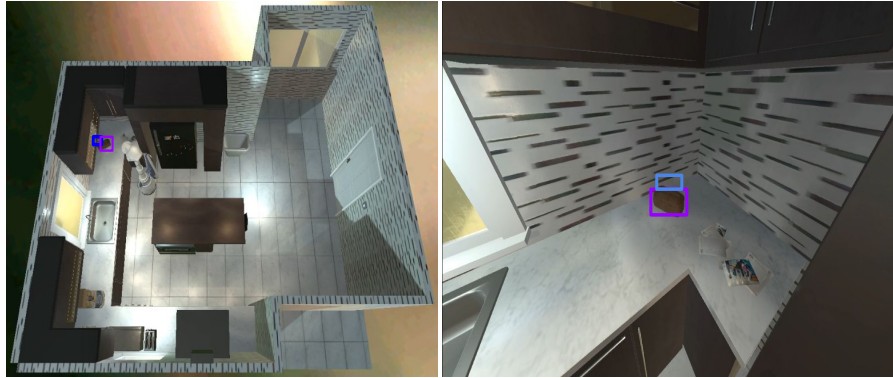

Figure 15: Occlusion of a fork(blue) by bread(pink) on the kitchen countertop.

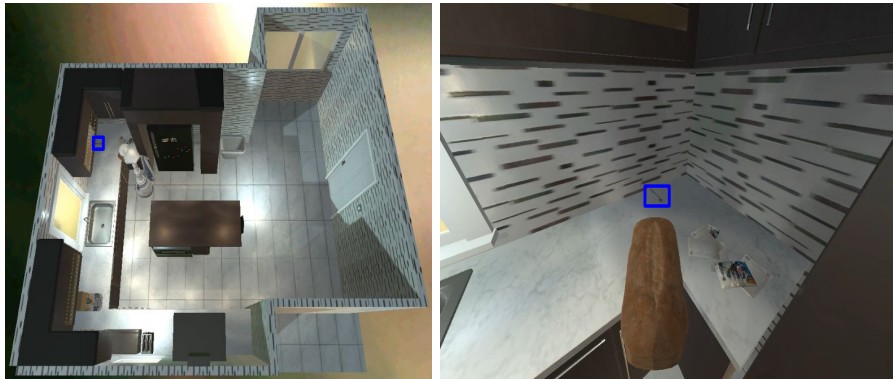

Figure 16: Occluded fork(blue) becomes visible when the bread(pink) is picked from the kitchen countertop.

- **Step 2:** We sample batch $\{s, a, \bar{s}\}$ from replay buffer and query the proxy network to produce $r_{predicted}$. We use $\{s, a, r_{predicted}, \bar{s}\}$ to update the Q-value by minimizing the Bellman error. We set the value of the learning rate for $\Phi_Q$ and $\Phi_p$ as 0.0001 and 0.000001 respectively. We use Adam optimizer for both networks.
- **Step 3:** We use polyak averaging to update the target networks of $\Phi_Q$ and $\Phi_p$. The value of the rate of averaging for target networks of $\Phi_Q$ and $\Phi_P$ is 0.0075 and 0.00085 respectively. We use an eviction policy similar to Kalashnikov et al. (2018) for our replay buffer.

We use PyTorch to train our models.

## I  QUALITATIVE RESULTS

We show the results of our method to solve the room-rearrangement problem under partial observability and swap cases with 5, and 10 objects in the supplementary video.

## J  ABLATION

We ablate our task planner against ground-truth perception (Ours-GT), various methods for object search (Ours-GE and Ours-RS) and a dense reward structure (Ours-DR). To study the effect of erroneous perception on our task planner, we assume the availability of **G**round-**T**ruth object detection labelling and 3D centroid localization from Ai2Thor Kolve et al. (2017). To understand the importance of Search Network in planning, we replace it by a **(i) R**andom **S**earch policy, which predicts probable receptacles for unseen objects with uniform probability and a **(ii) G**reedy **E**xploration strategy Chaplot et al. (2020) that optimizes for map coverage to discover all the unseen objects. To highlight the generalization of proxy reward network to the overall objective of the rearrangement episode, we

| Number of Objects | Visible Objects | Unseen Objects | | Swap Case | Ours-GT | | | Ours | | | Ours-RS | | | Ours-GE | | | Ours-DR | | |
|---|---|---|---|---|---|---|---|---|---|---|---|---|---|---|---|---|---|---|---|
| | | OOF | OPR | | SNS↑ | ENR↑ | ATC(m) | SNS↑ | ENR↑ | ATC(m) | SNS↑ | ENR↑ | ATC(m) | SNS↑ | ENR↑ | ATC(m) | SNS↑ | ENR↑ | ATC(m) |
| | 5 | 0 | 0 | 0 | 0.98 | NC | 10.57 | 0.74 | NC | 11.98 | 0.74 | NC | 11.98 | 0.74 | NC | 11.98 | 0.74 | NC | 12.76 |
| 5 | 5 | 0 | 0 | 2 | 0.70 | NC | 12.36 | 0.53 | NC | 13.46 | 0.53 | NC | 13.46 | 0.53 | NC | 13.46 | 0.53 | NC | 15.36 |
| | 3 | 2 | 0 | 0 | 0.81 | 0.61 | 12.93 | 0.60 | 0.48 | 14.33 | 0.46 | 0.30 | 22.23 | 0.36 | 0.20 | 26.38 | 0.60 | 0.48 | 16.18 |
| | 3 | 0 | 2 | 0 | 0.79 | 0.60 | 13.39 | 0.58 | 0.47 | 14.89 | 0.41 | 0.22 | 23.86 | 0.0 | NC | NC | 0.58 | 0.47 | 16.93 |
| | 10 | 0 | 0 | 0 | 0.97 | NC | 22.19 | 0.73 | NC | 24.51 | 0.73 | NC | 24.51 | 0.73 | NC | 24.51 | 0.73 | NC | 26.37 |
| 10 | 10 | 0 | 0 | 4 | 0.70 | NC | 24.63 | 0.52 | NC | 27.32 | 0.52 | NC | 27.32 | 0.52 | NC | 27.32 | 0.52 | NC | 29.46 |
| | 6 | 4 | 0 | 0 | 0.84 | 0.69 | 23.78 | 0.64 | 0.53 | 25.56 | 0.48 | 0.34 | 36.72 | 0.41 | 0.23 | 40.09 | 0.64 | 0.53 | 26.13 |
| | 6 | 0 | 4 | 0 | 0.83 | 0.67 | 24.15 | 0.62 | 0.52 | 25.97 | 0.43 | 0.25 | 38.47 | 0 | NC | NC | 0.62 | 0.52 | 26.59 |
| | 20 | 0 | 0 | 0 | 0.95 | NC | 40.05 | 0.73 | NC | 44.05 | 0.73 | NC | 44.05 | 0.73 | NC | 44.05 | 0.73 | NC | 48.27 |
| 20 | 20 | 0 | 0 | 8 | 0.70 | NC | 45.32 | 0.52 | NC | 48.32 | 0.52 | NC | 48.32 | 0.52 | NC | 48.32 | 0.52 | NC | 51.55 |
| | 12 | 8 | 0 | 0 | 0.87 | 0.75 | 41.29 | 0.67 | 0.58 | 45.29 | 0.51 | 0.36 | 52.45 | 0.46 | 0.28 | 56.68 | 0.67 | 0.58 | 47.42 |
| | 12 | 0 | 8 | 0 | 0.87 | 0.74 | 42.13 | 0.66 | 0.57 | 45.78 | 0.47 | 0.28 | 54.68 | 0 | NC | NC | 0.66 | 0.57 | 47.67 |

Table 3: (**OOF** : Objects outside agent's field of view initially, which are visible from a different perspective, **OPR** : Objects placed inside closed receptacles, **NC** : Not computable). Ours-GE fails to handle the partial observability due to **OPR** (**SNS = 0**). Whereas, Ours-RS addresses both the cases of partial observability but incurs a huge traversal cost (higher **ATC**) and takes more number of steps to search object (lower **ENR**) compared to Ours due to a random search policy. Ours-GE shows a higher **ATC** and lower **ENR** for **OOF** compared to both Ours and Ours-RS due to explicit exploration policy for search. Ours-DR shows a drop in **ATC** as compared to Ours due to the greedy planning policy based on the nearest neighbour reward.

| Number of Objects | Unseen Objects | | With SRTN | | | Without SRTN | | |
|---|---|---|---|---|---|---|---|---|
| | OOF | OPR | SNS↑ | ENR↑ | ATC(m) | SNS↑ | ENR↑ | ATC(m) |
| | 2 | 0 | 0.60 | 0.48 | 14.33 | 0.58 | 0.44 | 17.27 |
| 5 | 0 | 2 | 0.58 | 0.47 | 14.89 | 0.54 | 0.41 | 18.65 |
| | 4 | 0 | 0.64 | 0.53 | 25.56 | 0.60 | 0.47 | 29.35 |
| 10 | 0 | 4 | 0.62 | 0.52 | 25.97 | 0.56 | 0.45 | 30.94 |

Table 4: Performance of our Search Network with and without the Sorting Network (SRTN)

replace it with a hierarchical **D**ense **R**eward structure as in Ghosh *et al.*. Tab. 3 shows the results of the aforementioned ablation methods.

Without erroneous perception, *Ours-GT* demonstrates efficient planning, by performing significantly better than all the ablation methods in terms of **SNR**, **ENR** and **ATC**.

**Under complete visibility and swap cases** - Ours, Ours - RS and Ours - GE show similar results, since no partial observability cases exist. However, Ours-DR has a higher traversal (**ATC**) cost due to its greedy action selection based on the nearest neighbour reward. In contrast, Ours uses the episodic notion based proxy reward that considers the overall notion of the episode to train the Deep RL, which minimizes the agent's traversal.

**Under partial observability**, Ours performs significantly better than Ours-GE, Ours-RS and Ours-DR in terms **SNS**, **ENR** and **ATC**. This is due to the efficacy of the Search Network and the efficient planning of the Deep RL for simultaneous object search and rearrangement, which is trained with episodic notion based proxy reward. Whereas, Ours-GE incurs a high traversal cost in terms of **ATC** because it explicitly explores the entire room to find the **OOF** objects. Moreover, Ours-GE fails to address the **OPR** cases (**SNS = 0**) because the greedy exploration policy Chaplot et al. (2020) in terms of map coverage does not include opening and closing receptacles to find **OPR**. However, Ours-RS randomly visits receptacles to discover **OOF** or **OPR** cases, which again increases **ATC**. More number of attempts to search in Ours-GE and Ours-RS leads to a lower **ENR** as well as **SNS**. We observe that Ours-RS performs slightly better than Ours-GE in terms of **ENR** and **ATC** for **OOF**, because Ours-RS interleaves object search and rearrangement, rather than doing an explicit exploration strategy for finding objects. This is due to the fact that rearrangement of visible objects and search for some unseen objects, aids in the discovery of other unseen objects. Ours-DR shows a slightly higher **ATC** compared to Ours for both **OOF** and **OPR** cases in partial observability due to the greedy planning policy based on the nearest neighbour reward.

Further ablation study highlights the efficacy of different modules and our choice of hyper-parameters.

| Number of Objects | Unseen Objects | | With SCN | | | Without SCN | | |
|---|---|---|---|---|---|---|---|---|
| | OOF | OPR | SNS↑ | ENR↑ | ATC(m) | SNS↑ | ENR↑ | ATC(m) |
| 5 | 2 | 0 | **0.60** | **0.48** | **14.33** | 0.52 | 0.41 | 19.57 |
| | 0 | 2 | **0.58** | **0.47** | **14.89** | 0.49 | 0.38 | 20.83 |
| 10 | 4 | 0 | **0.64** | **0.53** | **25.56** | 0.54 | 0.45 | 31.17 |
| | 0 | 4 | **0.62** | **0.52** | **25.97** | 0.52 | 0.39 | 32.11 |

Table 5: Performance of our Search Network with and without the Scoring Network (SCN)

## J.1 SEARCH NETWORK

### J.1.1 IMPORTANCE OF SORTING NETWORK

In the two-stage approach, Sorting Network (SRTN) reduces the likelihood of ranking an implausible class room-receptacle in Scoring Network (SCN), thereby preventing inefficiencies in the search process. Tab. 4 shows the result of our search with and without SRTN. The slight degradation in performace in terms of SNS, ENR and ATC shows that without SRTN the method requires slightly more steps to search objects due to incorrectly scoring implausible class receptacles.

### J.1.2 IMPORTANCE OF SCORING NETWORK

The sorting network just classifies room-receptacles for objects into three groups based on the likelihood of finding the object at each location. However, it does not assign scores to rank them, consequently failing to prioritize the search in the most probable receptacle first. Without the scoring network, the search strategy must randomly select a room-receptacle from the higher probability class. This random selection degrades efficiency in terms of ENR and ATC, increasing the number of search attempts, as indicated in Tab. 5.

### J.1.3 REASON FOR ROBERTA FINETUNING

Large Language Models (LLMs) such as RoBERTa Liu et al. (2019) are trained on a huge corpus of available online text, enabling them to comprehend and make predictions based on general language understanding. Moreover, their large training data corpus enhances their ability to generalize to a diverse range of input data. However, it does not necessarily translate to optimal performance for specific tasks, thereby emphasizing the importance of fine-tuning. Fine-tuning serves as the key to harness full potential of these models, allowing it to proficiently grasp domain-specific terminology and generate tailored predictions. To illustrate, consider the task of predicting probable room-receptacles for unseen objects. In this context, possessing knowledge of the specific names of objects and room-receptacles within the environment becomes essential. Fine-tuning empowers the model to incorporate this domain-specific knowledge, allowing it to make more precise predictions and enhance its suitability for practical applications.

Suppose we need to find spoon, an unseen object, in an untidy living-room. For a pre-trained RoBERTa, consider the following masked prompts : **(i)** *In an untidy living-room, a spoon is usually placed in the <mask>* and **(ii)** *In an untidy living-room, a spoon is usually placed on the <mask>*. The output obtained by unmasking these prompts in decreasing order of score is : *middle, floor, table, corner, center, counter, carpet*. Whereas, the output from our Search Network module based on fine-tuned RoBERTa embeddings in decreasing order of score is : *table, carpet, coffee-table, console-table, bottom-cabinet, stool, sofa-chair*. As evident, the output produced by the pre-trained RoBERTa model is linguistically sound but fails to effectively address the specific problem at hand and lacks the domain knowledge of room receptacles within a household. In our evaluation on the test set of Search Network Dataset (Sec 3.1), the Search Network module based on fine-tuned RoBERTa embeddings exhibits approximately a 40% improvement in performance compared to using just the pre-trained RoBERTa model.

### J.1.4 DIFFERENT HIDDEN-LAYER ROBERTA EMBEDDINGS

We present quantitative results obtained by utilizing different embeddings from the RoBERTa-large model for the Filter and Ranking networks in Tab 6. These findings indicate that leveraging embeddings from the second-to-last hidden layer offers improved performance in capturing the underlying embedded commonsense within our Search Network framework.

| Input Embedding | %Accuracy ↑ | | | MSE ↓ | | |
|---|---|---|---|---|---|---|
| | **train** | **val** | **test** | **train** | **val** | **test** |
| Average-$2^{nd}$-last hidden layer | 92.87% | 81.67% | 79.05% | 0.001 | 0.046 | 0.031 |
| Average-last hidden layer | 88.41% | 59.19% | 57.93% | 0.01 | 0.04 | 0.09 |

Table 6: This table shows our ablation study against different input embeddings from Roberta-large for the Search Network. We tried the last and the second last hidden layer from RoBERTa and our results show that the second-last hidden layer better captured the commonsense knowledge.

| Edge feature of graph | SNS ↑ | ENR ↑ | ATC(m) ↓ |
|---|---|---|---|
| Euclidean distance | 0.73 | 0.71 | 29.78 |
| path length distance | 0.73 | 0.71 | 23.21 |

Table 7: Results to show the effect of using the path length distance based edge feature over the euclidean distance one.

## J.2 GRAPH REPRESENTATION NETWORK

We try to understand the effect of pre-trained GRN features on our state space. Fig. 13 shows that our pretrained GRN produces good features which capture the relative pairwise distance between the source and goal graph well. Further, we also show the effect of using the edge features of pairwise Euclidean distance v/s pairwise path length distance between the objects. The results in Tab. 7 show that the performance of the task planner is impacted significantly by using the edge feature as path length distance over the Euclidean one because the path length metric better captures the real scene geometry compared to the Euclidean.

## J.3 DEEP RL

We use the ablation study to decide the effective value of $\epsilon$. Fig. 11 shows that our Deep RL method works best for $\epsilon = 0.025$. Moreover, we have shown the effect of number of clusters for Cluster Biased Return Decomposition (CB-RD) in terms of sampling efficiency. Fig. 12 shows that the cluster size of three gives the best performance in terms of sampling efficiency.

## J.4 REWARD

We show the ablations for all the reward components mentioned in Appendix D in our supplementary video. The study highlights the significance of each reward component and also the improvement in the distance traversed due to the change in the Collision Resolution reward.

## K FAILURE CASE

As depicted in Fig. 15, there are instances where an object at the receptacle is hidden from the agent's egocentric view of the current state due to occlusion by another object. Our method can not resolve the search in such scenarios. To deal with these cases, we need a method that generates a set of manipulation actions such as pick-place and moves the objects which are causing the occlusion as shown in Fig. 16.

| High-level object categories | Objects |
|---|---|
| Cooking accessories | spatula, pot, kettle, pan |
| Kitchen Accessories | aluminum foil, bottle, , dish sponge, salt shaker, pepper shaker, ladle, soap bottle, scrub brush |
| Cutlery | butterKnife, knife, fork, spoon, plate |
| Crockery | bowl, cup, mug |
| Edibles | bread, bread-sliced, apple, apple-sliced, egg, egg-cracked, tomato, potato, lettuce, wine, wine-bottle, tomato-sliced, potato-sliced, lettuce-sliced |
| Sports Accessories | baseball bat, basketball, boots, tennis racket, dumbbell |
| Stationery | book, pen, pencil |
| Room-decor | watch, statue, vase, teddy-bear, table-top-decor, poster, pillow, painting, desk-lamp, floor-lamp |
| Bathroom accessories | towel, toilet-paper, soap-bottle, soap-bar, scrub-brush, plunger, paper-towel-roll, handtowel |
| Utilities | watering-can, vacuum-cleaner, tissue-box, spray-bottle, candle, mirror, alarm-clock, box |
| Consumer items | cell-phone, CD, remote-control, newspaper, laptop, credit-card, keychain |

Table 8: Table showing an example of objects in each of the high-level object categories.

