# OpenReview forum: "Task Planning for Visual Room Rearrangement under Partial Observability"
_ICLR.cc/2024/Conference — ICLR 2024 poster_

### Official Review · Reviewer_eRZf · 2023-10-15

**Soundness:** 2 fair
**Presentation:** 2 fair
**Contribution:** 2 fair
**Rating:** 6
**Confidence:** 4

**Summary:**

The paper proposes a hierarchical task planner equipped with several proposed components for room rearrangement for user-defined goal states.
Search Network exploits LLMs to query possible receptacles where unseen objects may be present.
The graph-based state representation encodes the objects' spatial relationships and their distances for the current and goal states in the form of graphs.
This is later used for Deep RL based Planner trained by the proposed cluster-biased return reward decomposition.
For the evaluation, the paper introduces a new benchmark, RoPOR, for room rearrangement that addresses blocked goals and swap cases, with newly introduced metrics that mainly measure agents' efficiency.
The proposed method outperforms the baselines in their empirical validations by noticeable margins.

**Strengths:**

- Tackling the blocked goals and swap cases is well-motivated and sounds sensible. Addressing them seems to be an important problem.
- Exploiting prior knowledge encoded in LLMs for possible target receptacles looks reasonable.
- Exploring an end-to-end framework for room rearrangement for user-defined goal states is intriguing.

**Weaknesses:**

- In Search Network, it is unclear why we need the "two-staged" approach: 1) filter out some implausible receptacles and 2) obtain the most plausible one. Why not just use SCN alone to get the most plausible receptacle, as the implausible receptacles should result in low scores and thereby be not chosen, consequently?
- The graph-based state representation requires shortest-path computation for all fully connected edges, but this seems quite computationally heavy, especially when we have a large number of nodes, leading to a drastically increasing number of edges.
- The comparison with some baselines seems unfair. For example, Weihs et al. and Gadre et al. do not use depth maps as input while the proposed method does, but they are compared in a single table. In addition, the authors utilize additional training datasets (Sec. 3.1).
- Some new metrics are introduced to measure agents' efficiency but they look a bit similar to SPL in navigation literature, which basically penalizes an agent's success rate by the length of trajectories it took so far. Similarly to the introduced metrics, as agents take more steps for rearrangement or search, SPL penalizes the success rates more, accordingly.
- The environments used in the proposed benchmark look quite "clean." It seems that we have objects only related to rearrangement tasks, as illustrated in Figure 16 and the supplementary video. This looks quite far from practical scenarios as we usually have many objects inside rooms.

\* Minor
 - It might be better to divide a result table (e.g., Table 1) into one for the main results and the other for the ablation study for better readability.

**Questions:**

See weaknesses above.

---

> ### Author Response · Authors · 2023-11-13
> **Response to Reviewer eRZf**
>
> **W1 :** In the two-stage approach, Sorting Network (SRTN) reduces the likelihood of ranking an implausible class room-receptacle in Scoring Network (SCN), thereby preventing inefficiencies in the search process. The ablation table below shows the result of our search with and without SRTN. The slight degradation in performace in terms of SNS, ENR and ATC shows that *without SRTN* the method requires slightly more steps to search objects due to incorrectly scoring implausible class receptacles.
>
> | Number of objects | Unseen |Objects||With SRTN|||| Without SRTN  ||
> |:----: | :-:| :-: |    -:  |   :-:  |   :-  |-|   -:  | :-:  |   :-  |
> |     |  OOF  | OPR| SNS&#8593;| ENR&#8593;| ATC&#8595;|| SNS&#8593;| ENR&#8593;| ATC&#8595;|
> |     5      |     2    |      0   |     0.60   |      0.48   |       14.33         | |   0.58  |      0.44     |     17.27     |
> |             |     0    |      2   |     0.58   |      0.47   |       14.89         | |   0.54  |      0.41     |     18.65     |
> |     10    |     4    |      0   |     0.64   |      0.53   |       25.56         | |   0.60  |      0.47     |     29.35     |
> |             |     0    |      4   |     0.62   |      0.52   |       25.97         | |   0.56  |      0.45     |     30.94     |
>
> **W2 :** Our 2D occupancy map, with average dimensions of 24x13 grids, where each grid represents 0.25m, yields an average time complexity of approximately $3\times10^7 \approx 0.031s$ even with N = 100 objects, for computing all pairs' shortest paths using Dijkstra's algorithm. This time complexity is expressed as $\mathcal{O}(N \times N \times (V+E\times logV))$, where vertices V = 24x13 and edges E = 4x(24x13-24-13).
>
> **W3 :** Similar to Trabucco *et al.* and Sarch *et al.*, depth map aids the perception stack of rearrangement to facilitate unique object identification through 3D centroids and anomaly detection between the goal and the current state. To achieve this, Weihs *et al.* and Gadre *et al.* employ different approaches based on image attention mechanism and continuous scene graph representation respectively. As these methods lack geometric information of objects, they fail to perform efficient planning. While Weihs *et al.* and Gadre *et al.* primarily focus on the perception stack for anomaly detection, our paper aims to address another fundamental problem in rearrangement, i.e. efficient task planning. In our table, we highlight the importance of efficient planning by comparing with these prior works, demonstrating that their lack of focus on efficient planning, results in a significant increase in traversal costs.
>
> The training datasets in Sec 3.1 are based on the train set of RoPOR and contribute to our objective of efficient task planning - **(i)** Search Network dataset helps us to train the semantic priors of human preferences in the untidy scenes, similar to the object-room-receptacle distribution in the train set of RoPOR. **(ii)** Graph dataset helps us to train the graph embedding to capture the geometric distribution of objects in the scene, similar to the train set in RoPOR. To ensure fair comparison, all the previous works were trained on the train set of RoPOR before performing evaluation in Table 1.
>
> **W4 :** Although SPL, SNS and ENR may seem similar in design, but serve distinct purposes and cater to a different problem. SPL metric is generally used in ObjectNav tasks, primarily penalizing the success rate of method for taking a longer path length. Whereas our SNS metric penalizes the agent for taking additional pick-place actions or search attempts in successful rearrangement scenarios with the same number of objects. ENR highlights the efficacy of our method in discovering unseen objects to overcome partial observability. ENR is simply the ratio of Number of unseen objects initially to the number of steps required to search them.
>
> **W5 :** To ensure clarity in the video illustration, we intentionally created clean scenarios. Typically, the RoPOR benchmark dataset generates scenes with many objects, but only a few of them require rearrangement.
>
> **W6 :** Table 1 presents a comparative study with prior works on RoPOR, while Table 2 provides a similar comparison on RoomR. Additionally, Table 3 showcases the ablation study conducted on RoPOR.

---

> > ### Author Response · Authors · 2023-11-21
> > **Requesting A Discussion, Reviewer eRZf**
> >
> > Please let us know whether our response has clarified any of your concerns. If not, please let us know what is not resolved and we can clarify those concerns. We look forward to have a discussion.

---

> > > ### Comment · Reviewer_eRZf · 2023-11-22
> > > **Official Comment by Reviewer eRZf**
> > >
> > > I thank the authors' detailed response with additional experiments.
> > > While the provided response addressed many of the raised concerns, some still remain unclear.
> > >
> > > > W3 : Similar to Trabucco et al. and Sarch et al., depth map aids the perception stack of rearrangement to facilitate unique object identification through 3D centroids and anomaly detection between the goal and the current state. To achieve this, Weihs et al. and Gadre et al. employ different approaches based on image attention mechanism and continuous scene graph representation respectively. As these methods lack geometric information of objects, they fail to perform efficient planning. While Weihs et al. and Gadre et al. primarily focus on the perception stack for anomaly detection, our paper aims to address another fundamental problem in rearrangement, i.e. efficient task planning. In our table, we highlight the importance of efficient planning by comparing with these prior works, demonstrating that their lack of focus on efficient planning, results in a significant increase in traversal costs.
> > >
> > > I understand how they work and focus on different aspects (here, anomaly detection), but it seems that the answer does not explain, so why the comparison with models without using depth maps (Weihs et al. and Gadre et al.) is fair.
> > >
> > > > W4 : Although SPL, SNS and ENR may seem similar in design, but serve distinct purposes and cater to a different problem. SPL metric is generally used in ObjectNav tasks, primarily penalizing the success rate of method for taking a longer path length. Whereas our SNS metric penalizes the agent for taking additional pick-place actions or search attempts in successful rearrangement scenarios with the same number of objects. ENR highlights the efficacy of our method in discovering unseen objects to overcome partial observability. ENR is simply the ratio of Number of unseen objects initially to the number of steps required to search them.
> > >
> > > The novelty of SNS still seems quite limited compared to SPL, as the same methodology of SPL is used for specific actions (here, pick and place). What are differences beyond the methodology?
> > >
> > > In addition, SPL can address what SNS does, because SPL also considers the additional pick-place actions. Can the authors provide some concrete examples where SNS works while SPL does not?

---

> > > > ### Author Response · Authors · 2023-11-22
> > > > **Response to Reviewer - eRZf**
> > > >
> > > > **W3:** Yes, we do understand that using an additional input in our pipeline seems unfair compared to Weihs *et al.*, 2021  and Gadre *et al.*, 2022. However, our evaluation table encompasses studies like Sarch *et al.*, 2022, and Ghosh *et al.*, 2022 that utilize depth maps in their methodologies. Furthermore, Sarch *et al.*, 2022 provides comparison results against Weihs *et al.*, 2022, and Gadre *et al.*, 2022 in their paper. Similarly, Ghosh *et al.*, 2022, provides comparisons with Weihs *et al.*, 2022. The rationale behind providing these evaluations is grounded in the common objective shared across these studies: addressing the challenge of room-rearrangement. Consequently, our method offers a comprehensive evaluation across various prior works, illustrating the chronological evolution in the research community's strategies to tackle the room-rearrangement problem.
> > > >
> > > > **W4:** *SPL* stands for Success measured by inverse of normalized path length cost - $S\frac{L}{max(P,L)}$. Here, $S$ denotes binary success criteria, $L$ is the shortest path length from the agent's initial position to the episode goal, and $P$ is the length of the path taken by the agent in that episode. This metric is impractical for rearrangement tasks due to the infeasibility of computing the length of the shortest possible path ($L$) needed to complete the rearrangement in an episode. Calculating $L$ involves a combinatorial complexity of $N!$, where $N$ is the number of objects in a rearrangement episode, making it computationally intractable for large number of objects.
> > > >
> > > > *SNS* stands for Success ($S$) measured by inverse of Number of steps ($N_T$) for a given number of objects ($N$) in a rearrangement episode - $S\frac{N}{N_T}$. Additionally, this metric signifies the efficiency of addressing scenarios involving a blocked goal or swap case. In a blocked goal scenario, an efficient task planner should adeptly rearrange $N$ objects without requiring any additional steps for buffer placement. Whereas, to resolve a swap case, an efficient rearrangement task planner should, at most, introduce an extra step for placing an object in a buffer space before moving it to the desired goal location. Hence, in the context of rearrangement, methods that require additional steps to address blocked goal and swap cases, even if they do not incur extra path length, should be considered inefficient. Apart from path length cost, the number of steps play a crucial role in evaluating the overall efficiency of rearrangement methods.
> > > >
> > > > Therefore, the *SNS* metric is valuable for assessing the overall rearrangement success and efficiency of future methods in dealing with blocked goal and swap cases with respect to the number of steps.

---

> > > > > ### Author Response · Authors · 2023-11-23
> > > > > **Requesting A Discussion, Reviewer eRZf - 2**
> > > > >
> > > > > Kindly inform us if our response has addressed your concerns. If there are still unresolved concerns, please specify, and we'll gladly provide further clarification.

---

> > > > > ### Comment · Reviewer_eRZf · 2023-11-23
> > > > > **Official Comment by Reviewer eRZf**
> > > > >
> > > > > I thank the authors for the clarification. For **W3**, I understand that they are included for a comprehensive evaluation, but still think that such additional input should be noted somewhere explicitly (*e.g.*, adding a column that indicates the usage of depth maps as input). For **W4**, it is now clear and I do not have any further questions about this.

---

> ### Author Response · Authors · 2023-11-23
> **Thank you, Reviewer eRZf**
>
> Thank you for dedicating your time and effort to review our paper and rebuttal. We sincerely appreciate your acknowledgment that our rebuttal has successfully addressed all your concerns. We will definitely incorporate the improvements stated in **W3**, that additional inputs such as depth are explicitly noted in a table column for the camera-ready version.
>
> *As our response has clarified your concerns, we kindly request you to consider updating your rating.*

---

### Official Review · Reviewer_JNsv · 2023-10-31

**Soundness:** 2 fair
**Presentation:** 2 fair
**Contribution:** 2 fair
**Rating:** 6
**Confidence:** 3

**Summary:**

The authors tackle the problem of object re-arrangement. To define the goal state, the agent is able to explore the room in its goal state and build up its own (graph-based) internal representation of the room. Then, at test time, the objects in the room are shuffled around such that they can be occluded or hidden in receptacles. The job of the agent is to put the room in the desired goal state as efficiently as possible. To do this, the proposed method keeps track of which objects have been seen and which haven't. A search network predicts probable locations of unseen objects. Both the current scene and goal scene are encoded as graphs, with objects as nodes and geodesic distances between objects as edges, and this scene is encoded with graph networks. Lastly, they use Q-learning with a proxy reward network to train the planner. To this end, the authors also build a dataset to train the search network and the graph network, in addition to contributing a new benchmark in Ai2Thor (RoPOR). The proposed approach significantly outperforms the baselines.

**Strengths:**

* The authors provide a new benchmark for object re-arrangement in AI2Thor. The new dataset supports swap cases (objects' positions that are swapped compared to the goal scenario) and blocked goals (ie. there is an object occupying the goal position of another object).
* Extensive supplementary video and supplementary material
* Modified RL training approach seems to outperform other approaches in that it converges substantially faster.
* The approach significantly outperforms the baselines.

**Weaknesses:**

* Convoluted approach that is a bit hard to follow, especially because many details are omitted in the main paper. It would also be helpful to more clearly define the task setup (inputs and outputs and their explicit representations).
* Not obvious why the SRTN needs to be learned instead of manually defined.
* Manual heuristics (where objects are unlikely to be found) and GT information (e.g. odometry, collision vector, etc.) is required.
* More details on the reward design in the main paper would be appreciated
* Critical ablations are not in the main paper.
* There is already an existing dataset (and associated metrics) for the same setup in Sarch et al. (Sec. 4.5 of their paper). While there are some deviations between the proposed dataset and the existing dataset, it still makes sense to benchmark the proposed approach against the already existing dataset that has baselines already benchmarked against it. Is there a reason this wasn't done? The setup of the existing seems like a subset of the proposed dataset's setup, so it seems doable.
* Somewhat similar to Sarch et al.
    * Both use 2D and 3D representations of the environment
    * Both use object detector + a search network to guide exploration
    * Both represent the scene as a graph on top of which they perform inference

Minor comments:
* Some strange wording such as contributing to the "research fraternity", misspelling such as "detetector, "Paramater", etc.
* Assumes perfect motion and manipulation
* Goal definition of having the goal scene already set up and utilizing an exploration stage is a bit impractical, as it requires setting up the goal scene every time it is changed. Also the robot needing to explore the goal state takes time compared to, say, language defined goals.

**Questions:**

* What are the failure cases? Does the agent ever get stuck in a loop?
* Are duplicate objects handled (e.g. multiple sponges in a scene)? Are these included in the test data, and how does the agent perform under these conditions?
* SRTN requires manually defined rules and heuristics about where objects are unlikely to be found (e.g. cup in bathtub). Is the network unable to learn these probabilities automatically?
* Does the SRTN need to be learned via the MLP? It seems like the probabilities can be pulled directly from the data without learning. We could simply build a hardcoded table of probabilities. For example, the probablity of finding a sponge in the sink can be hardcoded to 0.80. Is there a reason why that number would change? What is the benefit of learning here?
    * It is odd that we are trying to learn where objects are likely to be, but the training dataset is designed such that the authors "ensure a random distribution of object placements" (Appendix C.1).
* How does the agent go from the output of the planner to discrete motions (rotate, move forward, pick up, place)?
    * How is navigation performed when going from point A to point B? Is it assumed that this problem is solved?
* There is already an existing dataset (and associated metrics) for the same setup in Sarch et al. (Sec. 4.5 of their paper). While there are some deviations between the proposed dataset and the existing dataset, it still makes sense to benchmark the proposed approach against the already existing dataset that has baselines already benchmarked against it. Is there a reason this wasn't done? The setup of the existing seems like a subset of the proposed dataset's setup, so it seems doable.

---

> ### Author Response · Authors · 2023-11-13
> **Response to Reviewer JNsv**
>
> **W1 :** Task setup is described in Sec 2 introduction & Sec 2.1. To alleviate the reviewer's concern, we will improve the presentation in the camera-ready version.
>
> **W2 & Q4 :** The hand-labelled search's efficacy is limited to scenarios mentioned within it, whereas our Search Network module generalizes well to search unseen objects outside the training dataset due to the enormous semantic knowledge in LLM (shown by the seamless transferability to Habitat in supp video 19:17-20:26).
>
> **W3 :** No, manual heuristics are not required because the commonsense priors can be learnt from the train set of RoPOR. All prior works assume the GT information such as RGB-D and egomotion.
>
> **W4 & W5:** We will reconsider the space management in the main paper and shift the reward structure and critical ablations for the camera-ready version.
>
> **W6 & Q6 :** Yes, we have evaluated our method in existing dataset RoomR. Please find the comparative study in Table 2 and Sec B.
>
> **W7 :** Sarch *et al.* and our approach to room rearrangement is fundamentally different. They rely on general human commonsense in a tidy context to identify misplaced objects and utilize the visuo-semantic commonsense to locate a suitable receptacle for placing the object. However, their approach lacks consideration of geometric information, leading to inefficient planning as shown in Table 1. Whereas, we leverage the semantic knowledge in the untidy current state to search for unseen objects and use the geometric information to efficiently rearrange the visible misplaced ones, thereby minimizing the agent's overall traversal and number of rearrangement steps.
> - **W7.1 :** Yes, we both use 2D maps of the environment to aid navigation. For Sarch *et al.*, the 3D semantic map of the scene is required to find a suitable receptacle for placing a misplaced object according to commonsense of the tidy state. Whereas, we require the 3D map to augment the 3D centroids of the objects and receptacles to a global frame.
> - **W7.2 :** Both the methods use an object detector to understand the object configuration. Our Search network attempts to efficiently find the unseen object by leveraging human commonsense in the untidy state. Whereas, Sarch *et al.* aims to find suitable receptacles for placing the misplaced object, according to human commonsense for the tidy state.
> - **W7.3 :** Sarch et al. uses the scene graph and the Memex graph to identify misplaced objects and determine a suitable receptacle for placing them according to the commonsense knowledge of the tidy state. In contrast, our graph representation incorporates geometric information of objects to generate a graph embedding based RL state space. This ensures scalability and scene invariance.
>
> **W8 :** We will rectify them in the camera-ready version.
>
> **W9 :** This is a standard assumption in all the previous works.
>
> **W10 :** To perform rearrangement, the robot needs to know the tidy goal state, which is captured using a one-time goal state exploration. This approach is inline with state-of-the art methods. Yes, in the future we will study the integration of language-defined goal state, as it requires us to re-purpose the Search network to predict the location of objects from the language based goal, keeping our planning module intact.
>
> **Q1 :** In certain scenarios, the perception used in our method results in erroneous object detection. This leads to unnecessary search efforts and ineffective sequencing due to inaccuracies in object positions, causing failures in object pick/place actions. Another failure case arises when an unseen object is occluded by cluttered static objects - those that will not be moved during rearrangement.
>
> **Q2 :** Yes, duplicate cases are handled and are a part of the test data. In both the goal and current states, our object lists uniquely represent each object by incorporating their semantic label and 3D centroid position. Therefore, duplicate objects are treated the same as other objects and exhibit similar performance.
>
> **Q3 :** To learn automatically, the agent needs to explore all the untidy scenes in the train set of RoPOR and create a knowledge graph, which would then be used to finetune the LLM embeddings. This process is tedious and would not effectively capture all the plausible relationships, leading to poor generalizability. Therefore, we train SRTN to finetune the LLM embeddings using the manually defined data in the Search Network Dataset (SND), consisting of approximately 24,000 annotations where each object and room-receptacle pair is ranked by 10 individuals.
>
> **Q4.1 :** Refer the common response above.
>
> **Q5 :** Similar to all the previous works, we assume the availability of default navigation and motion planning. To generate a path from point A to B, we use the Djikstra's algorithm on the 2D occupancy map. Subsequent execution of low level action (such as Move forward or rotate right or pick-place) is performed using the built-in functions of Ai2Thor.

---

> ### Comment · Reviewer_JNsv · 2023-11-15
>
> Thank you for the thorough responses to the reviewers. The clarifications are helpful and it would be great if they could be clarified in the main paper, especially W2, W4, and Q1.
>
> **W1.** The task setup has some missing details. For example, it doesn't describe the action space or how the end of the episode is determined. How many objects can the agent hold at a time? What happens if the agent tries to pick up an object that isn't there? Can this happen?
>
> Please also improve the wording of these sections as they are written in an overly convoluted manner. For example, section 2.1 reads as having RGBD inputs, then using the RGBD inputs to create more inputs (M2D, M3D), then having an additional detector and its outputs as inputs, then having a set of other inputs (O, R, M2D, M3D) relating to the goal state, then mentioning that the agent has access to the detector again (note that it is misspelled as "detetector") that is used along with another input M3D to construct a final input O^V.
>
> An alternative way to describe this is that the agent uses as input: RGBD + egomotion, bounding boxes w/ 3D centroids, current state M2D, M3D, O_V, and goal state M2D, M3D, O. How each of these is computed can be described afterwards. Or, it can be mentioned that the agent only has access to RGBD + egomotion info from the current episode and from the entire goal exploration phase. It doesn't need to be written in exactly this way, but I find the current writing to be confusing. Similarly, we should have a clearly described action space.
>
> **Q5.** Is the info in the response written in the main paper? If not, it would be useful to add, as it connects the output of the planner to the action taken. Moreover, some additional questions are:
> * Is B computed from the detected object centroid?
> * How is the binary collision vector defined and how it is computed? Where does the robot get this from?
> * How does the planner output differentiate between searching a receptacle, picking up an object, and placing an object?
> * How is p_k defined? Is it a 3D location (centroid) from the object detector? How does it differ when selecting a receptacle vs picking up an object vs placing an object?

---

> > ### Author Response · Authors · 2023-11-15
> > **Response to Reviewer JNSv - 2**
> >
> > We appreciate the valuable feedback and the prompt response by the reviewer.
> >
> > **W1 :** In Section 2.4.1, the first paragraph details our hybrid action space, denoted as $a_i = (k, p_k)$. Here, $k$ represents the index of the chosen object, and $p_k$ indicates the 3D location for either object placement or receptacle search.
> >
> > The end of the episode is marked by the standard $done$ action from the Deep RL planner, which signifies the placement of all objects in the current state of rearrangement scenario to their goal location.
> >
> > The agent in Ai2Thor can hold only a single object at a time.
> >
> > No, the agent never tries to pick up an object that is not present. The planner chooses objects within the rearrangement scenario for pick-place. When the selected object is visible, the agent picks it up and places it in the goal location. In the case of an unseen object, the agent attempts to search the object at the predicted receptacle. If successful, the agent picks up the object and places it in its specified goal location. Conversely, if the agent fails to find an unseen object, it initiates a re-planning sequence as stated in Sec E.1.
> >
> > We will improve the ordering of the content in our task setup to incorporate the suggestions by the reviewer in our camera-ready version.
> >
> > **Q5 :** Yes, the response is present within the paper, last line of Sec 2.1.
> >
> > Yes, B is computed from the detected object centroid as stated in Sec 2.1.
> >
> > As stated in Sec E, the agent computes the binary collision vector ($C$), by identifying similarity between the current positions of visible objects with respect to the goal positions of other objects. The binary collision vector indicates the blocked and swap cases.
> >
> > As mentioned in Sec E, when $k$ points towards the visible misplaced object, the agent picks the object from its current location and places it in the goal location $p_k$. If $k$ points towards an unseen object, the agent attempts to search the object at the predicted receptacle position $p_k$. If successful, the agent picks up the object and places it in its specified goal location. Conversely, if the agent fails to find an unseen object at the receptacle position $p_k$, it initiates a re-planning sequence as stated in Sec E.1.
> >
> > As stated in Sec E, $p_k$ serves as a continuous parameter vector representing the 3D location for placing a visible object or searching an unseen one. For visible misplaced objects: **(i)** without swap case : $p_k$ indicates the placement position, aligning with the 3D goal centroid of the object; and **(ii)** with swap case : $p_k$ signifies the buffer position for placing the object to resolve swap case. For unseen objects, $p_k$ denotes the position of the predicted receptacle to search, derived from the 3D detection centroid of the predicted receptacle.

---

> > > ### Author Response · Authors · 2023-11-21
> > > **Requesting A Discussion, Reviewer JNsv**
> > >
> > > Please let us know whether our response has clarified any of your concerns. If not, please let us know what is not resolved and we can clarify those concerns. We look forward to have a discussion.

---

> > > > ### Comment · Reviewer_JNsv · 2023-11-21
> > > >
> > > > Yes, the response has clarified my concerns. Several of the clarifications are in the appendix — it would be appreciated if they could be moved into the main paper, though I understand there is limited space.

---

> ### Author Response · Authors · 2023-11-22
> **Thank you, Reviewer JNsv**
>
> We are grateful for the time and effort you dedicated to review both our paper and rebuttal. Your acknowledgment that our rebuttal has addressed all your concerns is humbly appreciated. We will diligently incorporate your suggestions into our camera-ready version.
>
> *As our response has clarified your concerns, we kindly request you to consider updating your rating.*

---

### Official Review · Reviewer_YNy7 · 2023-10-31

**Soundness:** 2 fair
**Presentation:** 2 fair
**Contribution:** 2 fair
**Rating:** 8
**Confidence:** 4

**Summary:**

This paper tackles the task of visual room rearrangement where an agent must first explore a 3D scene autonomously to map its content before objects in the scene are moved around. The agent must then, in a second phase, re-organize objects whose position has been changed compared with the initial scene. The authors propose a series of contributions to both evaluate methods, with a new benchmark and metrics, and improve autonomous agents’ overall performance and efficiency, with different architectural and training contributions.

First, the paper introduces a new Search Network whose goal is to predict the position of unseen objects after the room has been untidied. The claim is that such a search process could leverage prior common sense to find candidate receptacles more efficiently. Thus, the Search Network is composed of a large language model (LLM) that incorporates prior knowledge about the relationships between objects and receptacles.

Another contribution is a hybrid action space Deep RL agent that tackles both object search and rearrangement. The state space of this Deep RL agent is a graph representation of both the initial scene (known as goal state) and the current scene (known as current state) to provide information about the position of objects. Finally, the paper proposes a proxy reward network predicting a dense reward signal to facilitate the RL training of the policy.

The method is compared with different baselines on a new benchmark, RoPOR, and additional metrics, evaluating the efficiency of taken paths, are considered.

**Strengths:**

* **S1**: The paper tackles an important and challenging task, i.e. visual room rearrangement, and motivates its different contributions based on important considerations about the task.

* **S2**: This work proposes many different contributions, both from an evaluation point of view (benchmark, metrics) and from an architectural and training points of view.

* **S3**: The method is compared against different relevant baselines and shows a promising gain in performance.

* **S4**: Many qualitative videos are presented in the Supplementary Material, helping to better compare methods by visualizing their behavior.

**Weaknesses:**

* **W1**: **[Major]** How can the Search Network leverage prior knowledge as the room is untied? More specifically, if object shuffling/placement is done randomly, shouldn’t there be no prior remaining? Indeed, in phase 2, objects are placed in locations were there shouldn’t be (so that the agent can re-organize them). It is hard for me to understand how, in this case, the Search Network can learn anything meaningful. A result in the ablation study (Table 2) seems to confirm this intuition: the performance of Ours-RS, where the Search Network is replaced with a uniform random sampling of the next receptacle to visit, is very close (1p Success Rate) to the performance of Ours. The difference does not seem to be significant enough to claim the Search Network does more than random search. In order to claim a gain, authors should report the mean and standard deviation over a few training runs (random seeds). The following could also be done by authors:
    * **W1.1**: Reporting the performance of Ours-RS on the introduced RoPOR benchmark.
    * **W1.2**: Comparison of the Search Network with another simple baseline: selection of the closest receptacle (with reported performance on both RoPOR and RoomR).
    * **W1.3**: Provide more details about how objects are moved in phase 2: there should not be any prior remaining, and if there is, it might mean that the comparison is unfair with other methods because the authors’ search model might have been trained to learn those “shuffling priors” while it is probably not the case of previous work.

* **W2**: **[Major]** This comment is quite related to the previous one: as mentioned in the paper, the Search Network is finetuned to incorporate prior knowledge about object-receptacle relationships (see W1 about why I am not convinced any such relationship can be learned in the untidy scenario). Authors should still show the pre-training of the LLM brings a performance gain. What about the same LLM architecture initialized with random weights and trained as done in the paper?

* **W3**: **[Major]** It is not clear to me how the ground-truth data to train the Sorting Network (SRTN) is generated. Could authors elaborate on this?

* **W4**: **[Major]** One might argue that the Sorting Network only could be enough to predict the most likely object-receptacle pairs. Authors should provide an ablation study showing the impact of the additional Scoring Network.

* **W5**: **[Major]** I would like authors to clarify the following points regarding the Proxy Reward network:
    * **W5.1**: What is the interest of this Proxy Reward network? The paper mentions it is a way to predict a dense reward to train the RL agent. However, given a simulator, couldn’t we simply compute a dense reward from privileged simulator information at training time?
    * **W5.2**: What is the training ground truth for the Proxy Reward network?
    * **W5.3**: What is the average return on the y-axis of Figure 3? Such return is indeed associated with a specific reward function: what are the terms of this reward function? Moreover, I would like the authors to provide more details about the comparison done in Figure 3 and thus the conclusions we can draw from it.

* **W6**: **[Major]** The paper mentions the introduced method “assumes the availability of perfect motion planning and manipulation capabilities”. While this is a strong assumption, it does not outweigh the contributions in this work. However, an important question is: Are all the baselines this method is compared against also benefiting from the same assumption? Otherwise, this could be considered as an unfair comparison.

* **W7**: **[Minor]** When introducing their SNS metrics, authors should cite *Anderson et al., On Evaluation of Embodied Navigation Agents* that introduced quite similar metrics such as SPL.

* **W8**: **[Minor]** Paper citations are not properly inserted in the text. Authors should use parentheses (\citep{} in Latex) when needed, and remove double citations (e.g. “Sarch et al. Sarch et al. (2022)”).

**Questions:**

All questions and suggestions are already mentioned in the “Weaknesses” section as a list of numbered points.

---

> ### Author Response · Authors · 2023-11-13
> **Response for Reviewer YNy7**
>
> **W1 :** Refer to the response in the common concern section.
> Using SNS to gauge the search efficacy of Ours and Ours-RS is not entirely correct. Although, Ours-RS searches the receptacles uniformly, but it can eventually find the object through exhaustive search. To demonstrate this limitation, we have introduced two other metrics in the paper - ENR and ATC. ENR represents the ratio of initially unseen objects to the total search attempts needed to locate them. In Table 3 of the Appendix, the ENR values for Ours and Ours-RS demonstrate a widening gap, approximately 18% for 5 objects and increasing to about 22% for 20 objects. Moreover, the ATC difference between Ours and Ours-RS ranges approximately 30% to 50%. These metrics underscore the significance of search efficiency in Ours compared to Ours-RS.
> - **W1.1 :** Sec J and Table 3 in Appendix shows the performance of Ours - RS on the RoPOR dataset.
> - **W1.3 :** Refer the response in the common concern. For a fair comparison with state of the art methods, we have duly trained them in the train set of our benchmark RoPOR before evaluation. The benchmark dataset RoPOR is split into train, validation and test in the ratio of 70:10:20.
>
> **W2 :** Training an entire LLM module is a cumbersome process requiring vast amounts of textual data and huge computations. Instead, we leverage the principle of transfer learning by finetuning the LLM embeddings. Moreover, we have discussed the gain in performance resulting from the fine-tuning of LLM embeddings, compared to non-fine-tuned one. Please refer Sec J.1.1 of Appendix.
>
> **W3 :** We modify the AMT dataset to align with our objective, as stated in Sec 3.1 : Search Network Dataset (SND). Objects in AMT are split into training, validation, and test sets with a ratio of 70:10:20 in the SND dataset. Additionally, we compute an average normalized object-room-receptacle relationship score based on the mean inverse of the annotation rankings. The score's sign determines the ground-truth class label for room-receptacles of a given object, serving as the ground truth data for SRTN training : **(i)** Most probable (positive), **(ii)** Less probable (negative), and **(iii)** Implausible (zero).
>
> **W5.1 :** The motivation for proxy reward network is discussed comprehensively in the last paragraph of Sec 2.4.1 and Sec 2.4.2. Yes, we utilize a hierarchical dense reward structure, as stated in Sec D, to populate the replay buffer with step-wise reward for every episode. However, recent studies (Ren et al.) indicate that this handcrafted reward system performs well in scenarios within its intended design but struggles to generalize to extraneous cases. In addition, the dense reward structure provides feedback to the RL solely based on the outcome of each step, lacking consideration for the episodic objective. This leads to reduced sample-efficiency during RL training as shown in Fig 3 (paper). Therefore, we use proxy reward network to learn the reward distribution of the episodes from the experience data of the replay buffer, to give our agent a notion of the overall objective of the episode.
>
> **W5.2 :** The training ground truth for the proxy reward network is the scaled episodic reward of the randomly sampled episodes from the replay buffer. The episodic reward is the sum of rewards for all the steps in the episode. It is scaled with the ratio of Number of steps in the episode to the total path length cost for that episode, as mentioned in Eq 6. The training of the proxy reward occurs concurrently with the RL training, as explained in Sec 2.4.2 and Algorithm 1.
>
> **W5.3 :** Average return is the mean of rewards of the last 10 steps during RL training. We use a hierarchical dense reward structure, elucidated in Sec D of Appendix. In Figure 3, the number of training steps required by RL to consistently achieve a high average return is illustrated for different reward mechanisms, namely sparse, dense, proxy reward network trained with RRD, and CB-RD. It clearly establishes the sample-efficiency in training of our novel Cluster Biased - Return Decomposition (CB-RD) method, showcasing the highest return in the fewest training steps.
>
> **W6 :** Yes, these assumptions are in line with all the baselines.
>
> **W7 :** Sure, we will add this citation in the camera ready version.
>
> **W8 :** We will rectify this in the camera-ready version.

---

> ### Author Response · Authors · 2023-11-13
> **Experiments for Reviewer YNy7**
>
> **W1.2 :** The results for the baseline Ours-CR, employing the strategy of searching the unseen object in the nearest receptacle in both RoPOR and RoomR, reveals a notable resemblance in performance to Ours-RS. This is expected as both methods use heuristic search approaches—one random, the other prioritizing the closest receptacle—without incorporating commonsense. However, Ours substantially outperforms Ours-CR, particularly in terms of ENR and ATC. The success rates of Ours and Ours - CR are closely aligned in RoomR, as these metrics do not reflect the inefficiency in search based on the number of steps. The marginal decline in the success rate of Ours-CR is attributed to being timed out for the task, as it fails to search for objects efficiently.
>
> **Ours V/S Ours-CR in RoPOR**
> | Number of objects | Unseen |Objects||Ours|||| Ours-CR  ||
> |:----: | :-:| :-: |    -:  |   :-:  |   :-  |-|   -:  | :-:  |   :-  |
> |     |  OOF  | OPR| SNS&#8593;| ENR&#8593;| ATC&#8595;|| SNS&#8593;| ENR&#8593;| ATC&#8595;|
> |  5 |     2    |      0        |     0.60             |      0.48          |       14.33         | |      0.48          |         0.33       |     21.45           |
> |     |     0     |      2          |      0.58            |      0.47         |       14.89         | |       0.44          |          0.25      |         22.53       |
> |10 |     4     |      0         |      0.64            |      0.53          |       25.56         | |      0.52             |          0.37      |      34.38          |
> |     |     0     |      4         |      0.62          |      0.52           |       25.97           | |         0.47         |        0.31       |          36.65      |
>
> **Ours V/S Ours-CR in RoomR**
> | RoomR metrics |Ours| Ours-CR  |
> |:-----------: |   :----:  |   :----:  |
> |     Success Rate&#8593;      |    0.43             |      0.41         |
> |     Fixed Strict&#8593;      |     0.519     |      0.46        |
> |     Energy Remaining&#8595;    |     0.631     |      0.68       |
>
> **W4 :** The sorting network just classifies room-receptacles for objects into three groups based on the likelihood of finding the object at each location. However, it does not assign scores to rank them, consequently failing to prioritize the search in the most probable receptacle first. Without the scoring network, the search strategy must randomly select a room-receptacle from the higher probability class. This random selection degrades efficiency in terms of ENR and ATC, increasing the number of search attempts, as indicated in the ablation table below.
>
> **With SCN V/S Without SCN**
> | Number of objects | Unseen |Objects||With SCN|||| Without SCN  ||
> |:----: | :-:| :-: |    -:  |   :-:  |   :-  |-|   -:  | :-:  |   :-  |
> |     |  OOF  | OPR| SNS&#8593;| ENR&#8593;| ATC&#8595;|| SNS&#8593;| ENR&#8593;| ATC&#8595;|
> |     5      |     2    |      0        |     0.60             |      0.48          |       14.33         | |      0.52          |         0.41       |     19.57           |
> |             |     0     |      2          |      0.58            |      0.47         |       14.89         | |     0.49           |      0.38          |        20.83        |
> |     10      |     4     |      0         |      0.64            |      0.53          |       25.56         | |   0.54              |      0.45          |      31.17          |
> |             |     0     |      4         |      0.62          |      0.52           |       25.97           | |       0.52         |           0.39     |        32.11        |

---

> > ### Comment · Reviewer_YNy7 · 2023-11-21
> >
> > I thank the authors for their answers to my different concerns, along with additional experiments.
> >
> > My main remaining remarks are:
> >
> > - **R1**: the answer to **W1** (along with the common response) must be presented in more detail in the main paper as it is necessary to properly motivate the interest of the Search Network.
> >
> > - **R2**: The ablation studies about “Ours-CR” and “With SCN V/S Without SCN” should also be added to the main paper as they add significant value.
> >
> > I appreciate the clarifications from the authors and am thus increasing my overall score.

---

> > > ### Author Response · Authors · 2023-11-21
> > > **Thank you, Reviewer YNy7**
> > >
> > > We sincerely thank you for recognizing our efforts in addressing your concerns, leading to a positive change in your rating from 5 (Borderline Reject) to 8 (Accept). We assure you that the response to **W1** and the additional ablations showing the efficacy of our Search module (“Ours-CR” & “With SCN V/S Without SCN”) will be duly incorporated in the camera-ready version.

---

### Author Response · Authors · 2023-11-13
**Common response to all Reviewers**

We express our humble gratitude towards the reviewers for their valuable feedback and acknowledgment of our work's promising contributions, including the end-to-end rearrangement framework, a modified RL training approach with faster convergence, addressing blocked goal & swap cases, demonstrating significant improvement over baselines, and the introduction of a novel benchmark dataset for rearrangement in Ai2Thor.

**Common concern of Reviewer YNy7 & JNsv :** How can the Search Network leverage prior knowledge when the room is untidy, specifically when the distribution of objects during shuffling is random?\
**Ans.** Initially it might seem challenging to generalize an untidy scene, but prudent consideration reveals a discernible distribution. Despite the apparent chaos, human tendencies play a role in shaping the disorder, with specific locations being more likely for misplacing an item, reflecting the underlying principles of human behavior.

To create realistic scenarios during shuffling, object placement on the room-receptacles is not entirely random but follows certain semantic prior to reflect the underlying human preferences for untidiness. This semantic prior is induced using approximately 34,000 annotations, where each annotation comprises of rankings from 10 individuals, for each pair of 128 room-receptacles and 269 objects in the AMT human preference dataset (Kant *et al.*). A plausible room-receptacle for a given object is randomly selected based on the annotations. The Search Network dataset, follows a similar distribution of object-room-receptacles as the train set of RoPOR. During the training of the Search Network, we leverage this commensense prior by finetuning the LLM embedding to ensure generalizability to other datasets and environments, demonstrated by our method's performance in a new dataset RoomR (Weihs *et al.*) and seamless sim2sim transferability to Habitat (supplementary video 19:19 - 20:27).

---

### Meta-Review · Area_Chair_wdgi · 2023-12-03

**Metareview:**

The paper initially received borderline ratings, and the reviewers had several concerns regarding the clarity of the paper:
- Lack of clarity on how Search Network leverages prior knowledge
- Lack of clarity on the generation of groundtruth data for the Sorting Network
- Lack of clarity on what the Proxy Reward network does
- Lack of clarity on the two-staged approach of the Search Network
- And many more issues.

The rebuttal addressed the concerns of the reviewers. It also provided additional experiments for missing baselines and ablations. Hence, the reviewers increased their ratings.
The AC checked the paper, the reviews, the rebuttal, and the discussion. The paper tackles an interesting problem and achieves good results. So, acceptance is recommended. However, I strongly urge the authors to:

(1) Revise the paper according to the comments from the reviewers, and transfer the critical points from the appendix to the main paper.

(2) Release the code so others can reproduce the results.

Without these changes, the paper is going to be very weak.

**Justification For Why Not Higher Score:**

The paper requires revision due to numerous unclear details. While the rebuttal addresses the clarity issues, it is essential that the paper is revised for the camera-ready version. As it stands, the paper's current form falls short of meeting the criteria for a strong acceptance. Also, it is a highly engineered method for this specific task, and it is unlikely that it generalizes to other tasks.

**Justification For Why Not Lower Score:**

The paper tackles an interesting problem, provides a new dataset for the rearrangement task when the goal receptacles are blocked by other objects, and it achieves good performance.

---

### Decision · Program_Chairs · 2024-01-16

Accept (poster)